# Atmospheric data support a multi-decadal shift in the global methane budget towards natural tropical emissions

Alice Drinkwater[1,2], Paul I. Palmer[1,3], Liang Feng[1,3], Tim Arnold[2,1], Xin Lan[4,5], Sylvia E. Michel[6], Robert Parker[7,8], and Hartmut Boesch[7,8]

[1]School of GeoSciences, University of Edinburgh, Edinburgh, UK
[2]National Physical Laboratory, Teddington, UK
[3]National Centre for Earth Observation, University of Edinburgh, Edinburgh, UK
[4]Cooperative Institute for Research in Environmental Sciences, University of Colorado Boulder, Boulder, CO, USA
[5]Global Monitoring Laboratory, National Oceanic and Atmospheric Administration, Boulder, CO, USA
[6]Institute of Arctic and Alpine Research, University of Colorado Boulder, Boulder, CO, USA
[7]National Centre for Earth Observation, Space Park Leicester, University of Leicester, UK
[8]Earth Observation Science, School of Physics and Astronomy, University of Leicester

**Correspondence:** Paul I. Palmer (paul.palmer@ed.ac.uk)

**Abstract.** We use the GEOS-Chem global 3-D model and two inverse methods (the Maximum *A Posteriori* and Ensemble Kalman Filter) to infer regional methane ($CH_4$) emissions and the corresponding carbon stable isotope source signatures, 2004–2020, across the globe using *in situ* and satellite remote sensing data. We use the Siegel estimator to determine linear trends from the *in situ* data. Over our 17-year study period, we estimate a linear increase of 3.6 Tg/yr/yr in $CH_4$ emissions from tropical continental regions, including North Africa, southern Africa, tropical South America, and Tropical Asia. The second largest increase in $CH_4$ emissions over this period is from China (1.6 Tg/yr/yr). For Boreal regions we estimate a negative emission trend of -0.2 Tg/yr/yr and for northern and southern temperate regions we estimate trends of 0.03 Tg/yr/yr and 0.2 Tg/yr/yr, respectively. These increases in $CH_4$ emissions are accompanied by a progressively isotopically lighter atmospheric $\delta^{13}C$ signature over the tropics, particularly since 2012, which is consistent with an increased biogenic emission source and/or a decrease in a thermogenic/pyrogenic emission source that has a heavier isotopic signature. Previous studies have linked increased tropical biogenic emissions to increased continental rainfall, particularly over Eastern Africa. Over China, we find a weaker trend towards isotopically lighter $\delta^{13}C$ sources, suggesting that heavier isotopic source signatures play a larger contribution to this region. Satellite remote sensing data provide additional evidence of emission hotspots of $CH_4$ that are consistent with the location and seasonal timing of wetland emissions. The collective evidence suggests that increases in tropical $CH_4$ emissions are from biogenic sources, with a significant fraction from wetlands. To understand the influence of our results to changes in the hydroxyl radical (OH), we also report regional $CH_4$ emission estimates using an alternative scenario of a 0.5%/yr decrease in OH since 2004, followed by a larger 1.5% drop in 2020 during the first COVID-19 lockdown. We find that our main findings are broadly insensitive to those idealised year-to-year changes in OH, although the corresponding change in atmospheric $CH_4$ in 2020 is inconsistent with independent global-scale constraints for the estimated annual mean atmospheric growth rate.

# 1 Introduction

Changes in atmospheric methane ($CH_4$) over the last few decades have unfolded without clear explanation, exposing inadequacies in our measurement coverage and our ability to definitively attribute those changes to individual emissions and losses. The climatic importance of atmospheric $CH_4$ lies in its ability to absorb and emit infrared radiation at wavelengths that are relevant to outgoing terrestrial radiation and incoming shortwave radiation (Allen et al., 2023). Consequently, atmospheric $CH_4$ helps to maintain Earth's radiative balance and surface and atmospheric temperatures. Atmospheric $CH_4$ is derived from emissions due to thermogenic (organic matter broken down at high temperatures and pressures, mainly released during extraction and transport of fossil fuels), pyrogenic (through incomplete combustion of organic matter), and biogenic (microbial activity) based production pathways. The main loss process is from from the hydroxyl radical (OH), with minor losses from the reaction with chlorine, uptake from soils, and stratospheric loss. Methane is the second most abundant anthropogenic greenhouse gas in terms of its anthropogenic radiative forcing. The global $CH_4$ growth rate was close to zero from 2000 to 2006 (Dlugokencky et al., 2020) but has since accelerated, with a global annual growth rate reported by NOAA exceeding 15 ppb for the first time in 2020 and more than 18 ppb in 2021 (Feng et al., 2023). Concurrently, we are witnessing a progressively isotopically lighter signature of global averaged $CH_4$ (more negative global average atmospheric $\delta^{13}C$ value). Analysis of $CH_4$ mole fraction and $\delta^{13}C$-$CH_4$ data suggest that thermogenic sources are unlikely to be the dominant driver of the post-2006 global mean increase in atmospheric $CH_4$ (Lan et al., 2021)). A growing body of work has proposed a range of hypotheses to explain short periods of observed global and regional variations in atmospheric $CH_4$ (Turner et al., 2019). In this study, we take a step back to look at observed $CH_4$ variations from 2004 to 2020, in order to capture the some of the zero-growth rate period and the subsequent increase in growth rate of $CH_4$ post-2007. We argue that monthly variations are part of a large-scale shift of predominately thermogenic energy emissions from high northern latitudes to biogenic emissions from the tropics, driven by larger emissions over tropical North Africa and tropical South America.

The post-2007 increase in atmospheric $CH_4$ has been the focus of many studies and has been attributed to different plausible hypotheses associated with changes in various emissions sources, and the OH sink (Turner et al., 2019). These studies have reached their conclusions using *in situ* mole fraction observations alone or in combination with other observations, e.g. *in situ* $\delta^{13}C$ (Schaefer et al., 2016; Rice et al., 2016; Nisbet et al., 2016; Fujita et al., 2020; Lan et al., 2021; Basu et al., 2022; Oh et al., 2022), satellite observations (Worden et al., 2017; McNorton et al., 2018; Yin et al., 2021; Feng et al., 2022), or other trace gases, using a variety of analysis methods and computational models. Typical emissions sizes and uncertainty are indicated in Table 1, adapted from Saunois et al. (2020). Our approach is unique in that, for our $\delta^{13}C$ inversion, we are solving for the $\delta^{13}C$ isotopic source signature of a geographical region. From the isotopic source signature of a region, we can determine how the source balance within a particular region has shifted over time, e.g. larger or smaller contributions from pyrogenic and biogenic sources, and consequently gain understanding of the geographical shifts in the $CH_4$ budget.

Methane oxidation by the OH radical in the troposphere is responsible for 80% of the total $CH_4$ sink globally. Changes in OH may have played a role in recent changes in atmospheric $CH_4$ (Rigby et al., 2017; Turner et al., 2017) but the magnitude of this influence is uncertain (its short atmospheric lifetime of <1 s makes direct measurement of global variability very difficult).

Reducing values of OH, effectively increases atmospheric $CH_4$ and therefore has the same effect as increasing emissions of $CH_4$. Chemical reactions responsible for removing $CH_4$ from the atmosphere are faster for lighter isotopologues of $CH_4$. This isotopic fractionation therefore leads to an atmosphere enriched in heavier isotopes relative to the globally emitted $CH_4$. Lan et al. (2021) simulated $CH_4$ and $\delta^{13}C$ in a 3-D chemistry transport model covering the period 1984-2016, and found that changes in OH proposed by Turner et al. (2017) are not consistent with the trend of increasingly isotopically light $\delta^{13}C$ observed in the atmospheric record. We explore the impact of reducing OH in a sensitivity study, taking into account a larger OH decrease during 2020 (Peng et al., 2022; Feng et al., 2023) that was associated with widespread redutions in nitrogen oxide emissions (Cooper et al., 2022).

Here, we calculate trends in regional $CH_4$ emissions and isotopic $\delta^{13}C$ source signatures across the world, 2004–2020, using *in situ* mole fraction and $\delta^{13}C$ data, and satellite column-averaged dry-air mole fraction data. This is achieved by using three sets of inversions: two Maximum A-Posteriori inversions using ground-based data (solving separately for regional emissions and isotopic sources signatures), and an Ensemble Kalman Filter inversion using data from the Japanese Greenhouse gases Observing SATellite (GOSAT) that solves for regional $CH_4$ emissions.

In the next section, we describe the data and methods we use to quantify changes in regional $CH_4$ emissions and the corresponding regional stable isotope source signatures. In section 3, we report our results of *a posteriori* regional $CH_4$ fluxes and regional $\delta^{13}C$ isotopic signatures, including analysis of sensitivity calculations that involve different assumptions about year to year changes in the OH sink. We conclude the paper in section 4.

## 2   Data and Methods

### 2.1   *In Situ* and Satellite Remote Measurements of Atmospheric Methane

We use surface-level flask data as constraints on both regional $CH_4$ emissions and $\delta^{13}C$ regional $CH_4$ emissions isotopic source signatures. The $CH_4$ mole fraction data are taken from 31 National Oceanic and Atmosphere Administration – Global Monitoring Laboratory (NOAA-GML) sites around the world (Figure 1), version 2020-07 (Dlugokencky et al., 2020). The data are monthly mean values, averaged from discrete data as collected at each site, analysed at NOAA-ESRL in Boulder, Colorado, and recorded to the NOAA 2004A standard scale (Dlugokencky et al., 2005). Up to August 2019, the analysis was performed using gas chromatography (Steele et al., 1987, Dlugokency et al., 1994; Dlugokencky et al., 2005) and since August 2019, cavity ringdown spectroscopy has been used (Dlugokencky et al., 2020). We also include data from a site in Siberia, Karasevoe (KRS), which is monitored by the National Institute for Environment Studies (NIES). This site was included to maximise geographical coverage of *in situ* data. The $CH_4$ mole fraction measurements from this site are continuous, measuring from 65 m height, covering the period 2004–2020 (Sasakawa et al., 2010). A scale factor of 0.997 is applied to the NIES data in order to bring it into line with the NOAA 2004A scale (Zhou et al., 2009). The site constitutes part of the Japan-Russia Siberia Tall Tower Inland Observation Network (JR-STATION).

$\delta^{13}C$ data are similarly monthly mean values, calculated from discrete flask samples at NOAA network sites, reported on the international carbon isotope scale VPDB (Vienna Pee Dee Belemnite). Isotope ratio 'delta' values represent the excess

of a heavy, less abundant stable isotope (for $\delta^{13}C$ values, carbon-13) over the light, most abundant stable isotope (carbon-12) in a sample, when compared to a standard. These measurements are useful as they are indicative of the source of the

$CH_4$: biogenic sources are dominated by isotopically lighter signatures and thermogenic sources are dominated by isotopically heavier signatures. For the NOAA network, isotopic analysis of $\delta^{13}C$ was performed at the University of Colorado Institute of Arctic and Alpine Research Stable Isotope Laboratory (CU-INSTAAR). They follow an isotope ratio mass spectrometry approach (Miller, 2002; Vaughn et al., 2004). The geographical locations of *in situ* measurement sites are shown in Figure 1. These sites are a subset of the entire NOAA network's capacity for measuring $CH_4$ mole fractions. The sites included in the

inversion (both for $CH_4$ and $\delta^{13}C$) are those that cover the entire period of the inversion (2004-2020) without significant periods of measurement breaks to ensure a consistent interpretation of trends without consideration of possible biases introduced through the inclusion or exclusion of specific sites.

      We also estimate $CH_4$ fluxes for 2010-2020 using data from GOSAT that was launched in 2009. GOSAT is in a sun-synchronous orbit with an equatorial local overpass time of 13:30. Since launch, it has provided continuous global observations

of dry-air atmospheric column-averaged $CO_2$ ($XCO_2$) and $CH_4$ ($XCH_4$), retrieved from shortwave infrared wavelengths that are most sensitive to changes in $CH_4$ and $CO_2$ in the lower troposphere (Parker et al., 2020). We use the latest (v9) proxy $XCH_4$:$XCO_2$ retrievals that use spectral absorption features around the wavelength of 1.6 $\mu$m (Parker et al., 2020, Palmer et al., 2021), because of the smaller bias and better global coverage than those provided by the full physics retrievals. Analyses show the precision of single proxy retrieval is about 0.72%, with a global bias of 0.2% (Parker et al., 2011, 2015, 2020). In

our calculations, we assume a higher observation uncertainty of 1.2%, and deduct a globally uniform bias of 0.3% to obtain better *a posteriori* agreement with the independent ground-based $XCH_4$ data by the Total Carbon Column Observing Network (TCCON). These uncertainties are detailed in Feng et al. (2022). To anchor the constraints from the proxy $XCH_4$:$XCO_2$ ratio (Fraser et al., 2014; Feng et al., 2017), we also assimilate the GLOBALVIEW $CH_4$ and $CO_2$ data (Schuldt et al., 2021), with assumed uncertainties of 0.5 ppm and 8 ppb for *in situ* measurements of $CO_2$ and $CH_4$, respectively. GLOBALVIEW

constitutes a combination of $CH_4$ data from ground-based data (both flask and continuous) and aircraft data, from 54 different laboratories, combined and published by NOAA-GML (Schuldt et al., 2021). Locations of the assimilated GLOBALVIEW $CH_4$ (sub) dataset are shown in Feng et al., 2022.

## 2.2   GEOS-Chem Atmospheric Chemistry and Transport Model

To relate $CH_4$ emissions to atmospheric $CH_4$ concentrations, we use v12.1 of the GEOS-Chem 3-D global chemical transport

model (CTM) (Bey et al., 2001) at a horizontal resolution of 2° (latitude) by 2.5° (longitude) with 47 vertical levels from the surface to 80 km height, with meteorological data from the MERRA-2 meteorological reanalyses (Gelaro et al., 2017) from the NASA Global Modeling and Assimilation Office (GMAO).

      Our *a priori* emissions include: 1) monthly EDGAR v6 anthropogenic emissions (Crippa et al., 2021) that accounts for emissions from oil and gas, coal, livestock, landfills, wastewater, rice, and other anthropogenic sources (including biofuel)

from 2004 to 2018, after which we repeat 2018 emission estimates; 2) monthly GFED-4 biomass burning emissions (version 4.1; Randerson et al., 2017); and 3) monthly v1.0 WetCHARTs wetland emissions (Bloom et al., 2017). The Harvard-NASA

Emissions COmponent (HEMCO) software within GEOS-Chem converts the emission inventories at their native horizontal resolution to the GEOS-Chem $2° \times 2.5°$ resolution. Beyond the end of the emissions inventory, emissions are repeated yearly in *a priori* simulation.

Table 1 shows the $\delta^{13}C$ signatures for the source types included in our simulations. These are extracted as mean global values from Sherwood et al. (2017), which provide a database of global isotopic source signatures that are broken down into the same sectors as we employed in our simulations. However, individual source types show a wide range of source signatures, and this uncertainty is reflected in the assigned uncertainty given to the *a priori* source signatures in inversion (Section 2.3). In the inversion, we differentiate between Arctic and tropical wetlands by applying a 10‰ isotopically lighter source signature

to the Arctic source (Table 1), following Ganesan et al. (2018) who produced a global wetland source signature map based on published $\delta^{13}C$ data. Recent work showed that atmospheric simulations that included this isotopic distinction between Arctic and tropical wetlands provided clearer support for rising microbial emissions being responsible for a large fraction of the increase in atmospheric $CH_4$ since 2007 (Oh et al., 2022). In GEOS-Chem, we simulate isotopologues separately (i.e. for $\delta^{13}C$, $^{12}CH_4$ and $^{13}CH_4$), and then calculate $\delta^{13}C$ values. The arithmetic underlying the conversion of isotope ratios to

isotopologue emissions for input to the model are detailed in Appendix A.

     We include the loss of atmospheric $CH_4$ from reaction with chlorine, soil uptake, and from oxidation by OH. We use monthly 3-D fields of OH, calculated using the full-chemistry version of GEOS-Chem, and monthly 3-D field of atomic chlorine (Sherwen et al., 2016). Stratospheric loss frequency fields are determined using the NASA GMI stratospheric model (Duncan et al., 2007). Estimates of the microbial consumption of $CH_4$ in soils is determined from Fung et al. (1991). The

resulting atmospheric lifetime of CH4 against OH is 9.77 years. The corresponding lifetime for methyl chloroform is 5.41 years, which is consistent with atmospheric observation of methyl chloroform. This lifetime also compares well with multimode (Voulgarakis et al., 2013; Morgenstern et al., 2017) that reported global mean lifetimes of $CH_4$ that range 7.2–10.1 yrs. In our default model configuration, none of these loss processes include interannual variations.

     To account for isotopic fractionation due to loss of $CH_4$ in the troposphere and stratosphere, we use published kinetic

isotope effect values (KIEs). These values are employed to scale the reaction rate constants used in the simulations for $^{12}CH_4$ and $^{13}CH_4$ (Table A1). The OH and Cl sinks are handled in the hard coding of the model, whereas the soil sink is handled as a negative emission in the HEMCO file. Therefore, for the soil sink, the KIE is directly applied as a scale factor in the HEMCO configuration file (Snover and Quay, 2000; Burkholder et al., 2019).

     We created the initial conditions for atmospheric $CH_4$ by first scaling a standard $CH_4$ GEOS-Chem restart file (a file contain-

ing a default realistic distribution of $CH_4$ across the atmosphere) to conditions near representative of the start of our analysis in January 2004. We then ran the model sixty times with repeating 2004 MERRA-2 meteorology and emissions (corresponding to approximately six e-folding lifetimes for $CH_4$) to improve as far as possible the simulation of atmospheric gradients in CH4 in the initial conditions. We then ran a single-year inversion for 2004 to optimise the isotope ratios and $CH_4$ concentrations relative to ground-based observations, following the inverse method detailed below. The $\delta^{13}C$ inversion used the regional emis-

sions estimate provided by the posteriori from the $CH_4$ inversion as a starting point, with sectoral emissions scaled as detailed

in Appendix A. The output of this 2004 inversion is a final step in creation of the initial conditions, which serve as a starting point for the longer inversion that we report here (2004- 2020).

For all our calculations, we sample GEOS-Chem at the grid box and local time that corresponds to the *in situ* and satellite remote sensing data. For the satellite data, we also apply scene-dependent averaging kernels to account for vertical structure.
This approach allows us to directly compare the model with measurements. Regional trends are calculated by examining the grid boxes encompassed by a given region on the global grid.

## 2.3 Inverse Methods

We use two inverse methods that reflect the volume and simplicity of the data being used. For *in situ* data we use the Maximum *A Posteriori* (MAP) inverse methods and for the more voluminous satellite data we use an ensemble Kalman filter (EnKF). For
brevity, we include only the essential details about either method and refer the reader to dedicated papers.

### 2.3.1 Maximum *A Posteriori*

To infer regional *a posteriori* $CH_4$ fluxes and regional $\delta^{13}C$ emissions source signatures from the atmospheric measurements of $CH_4$, we use the Maximum *A Posteriori* solution (MAP) inverse method (Rodgers, 2000). We solve for $CH_4$ fluxes and $\delta^{13}C$ emissions signatures from 14 geographical regions (Figure 1). This method combines *a priori* knowledge and its uncertainty
with the measurements and their uncertainties, and has been used in a number of studies, e.g., Fraser et al. (2014); McNorton et al. (2018).

The MAP solution and the associated *a posteriori* uncertainty is described as, respectively:

$$\mathbf{x}^a \quad = \quad \mathbf{x}^b + (\mathbf{H}^T\mathbf{B}^{-1}\mathbf{H} + \mathbf{R}^{-1})^{-1}\mathbf{H}^T\mathbf{B}^{-1}(\mathbf{y} - \mathbf{H}\mathbf{x}^b), \tag{1}$$

$$\mathbf{A} \quad = \quad (\mathbf{H}^T\mathbf{B}^{-1}\mathbf{H} + \mathbf{R}^{-1})^{-1}, \tag{2}$$

using the conventional that lower-case and upper-case variables denote vectors and matrices, where $\mathbf{x}$ denotes the state vector that describes the estimated quantities, which in this study includes monthly $CH_4$ fluxes and $\delta^{13}C$ source signatures from regions across the world (Figure 1). Subscripts 'a' and 'b' denote *a posteriori* and *a priori* $CH_4$ fluxes, respectively, and superscripts '-1' and 'T' denote matrix inverse and transpose operations, respectively. The measurement vector $\mathbf{y}$ includes $CH_4$ mole fraction or $\delta^{13}C$ data. The matrices $\mathbf{B}$, $\mathbf{A}$, and $\mathbf{R}$ denote the error covariances matrices for the *a priori*, *a posteriori*, and
measurements, respectively. $\mathbf{B}$ and $\mathbf{R}$ are diagonal matrices. For $\mathbf{B}$ we assume uncertainties of 50% of the regional $CH_4$ fluxes and 15‰ for the $\delta^{13}C$ values, and for $\mathbf{R}$ we assume 10 ppb for the mole fraction data and 0.1‰ for the isotope data. These uncertainties were based on similar studies (Fraser et al., 2014; McNorton et al., 2016). We assume a model transport error of 12 ppb, following Feng et al. (2022).

The Jacobian matrix $\mathbf{H}$ describes the sensitivity of the measurements to changes in the state vector, i.e. $\partial \mathbf{y}/\partial \mathbf{x}$. For the mole
fraction $CH_4$ inversion, the Jacobian matrix describes the sensitivity of mole fractions in the model to changes in regional $CH_4$ emissions. We construct the matrix using a series of GEOS-Chem model runs. We systematically let each individual emitting region (described by the state vector) emit for one month while all other regions are emitting as normal. The individual regional

source is then switched off (emissions set to zero) and the effect of this on the 3-D atmospheric distribution of $CH_4$ mole fractions is recorded over the following three months. The result of this test is recorded at the grid boxes that correspond to the location of the measurement sites. The resulting mole fractions therefore describe the sensitivity of a particular measurement site to changes in a specific regional source up to three months after emission. This is repeated for every month within the inversion timescale, for every region described in the state vector.

For the $\delta^{13}C$ inversions, the Jacobian matrix describes the sensitivity of modelled $\delta^{13}C$ to changes in the regional isotopic source signatures. We construct the Jacobian as the difference between a control model calculation (using the $CH_4$ *a posteriori* regional emissions and mean source signature values from Sherwood et al. (2017)) and perturbed source signature model calculation for the whole study period (2004-2020). For the perturbed model calculation, we systematically perturb the isotopic source signature of each region (all of the sectors that are contained geographically within a region) isotopically heavier by 20‰ for the period 2004-2020. The difference between the control and perturbed run in $\delta^{13}C$ value at the location of each measurement site is then divided by the value of $\delta^{13}C$ perturbation for the region source signature, to understand the effect of changing a regions source signature on the $\delta^{13}C$ value recorded at each measurement site location.

The output from the inversion are improved estimates of regional $CH_4$ fluxes and $\delta^{13}C$ source signatures. The model simulates the global atmosphere on a $2° \times 2.5°$ horizontal grid. The *a posteriori* regional $CH_4$ fluxes and isotopic source signatures are applied to the grid boxes in the model which correspond to a given region in an *a posteriori* simulation.

### 2.3.2   Ensemble Kalman Filter

We use an Ensemble Kalman Filter (EnKF) approach in performing the inversion using satellite data, because we cannot easily evaluate the necessary matrix operations associated with an analytic inversion. Here we use an ensemble of flux perturbation pulses to represent uncertainty in our *a priori* estimate for regional monthly $CH_4$ fluxes. We subsequently use a global chemistry transport model (i.e., the GEOS-Chem v12) to track the transport and chemistry processes of the tagged emission pulses in the atmosphere, to project their spreads to the observation space. With the ensemble of *a priori* flux perturbations, and the simulated observation impacts, we use the Ensemble Transform Kalman Filter (ETKF) algorithm to numerically estimate the *a posteriori* $CH_4$ fluxes and the associated uncertainties by optimally comparing the model simulation with observations (see Feng et al., 2017 for more details). To reduce the computational costs, mainly from tracking tagged emission pulses, we introduce a 4-month moving lag window for each assimilation step, because any observation has limited ability to distinguish between the signals emitted long (>4 months) before, from variations in the ambient background atmosphere (Feng et al., 2017). As a result, we are able to include a larger state vector, consisting of monthly scaling factors for 487 (476 land regions and 11 oceanic regions) regional $CH_4$ (and $CO_2$) pulse-like basis functions (Figure S1 in (Feng et al., 2022)). We define these land sub-regions by dividing the 11 TransCom-3 (Gurney et al., 2002) land regions into 42 to 56 nearly equal sub-regions, and use the 11 oceanic regions defined by the TransCom-3 experiment. Because of their smaller sizes, we have assumed a higher uncertainty percentage (60%) for *a priori* emissions than the MAP approach described above. We also include spatial correlation with a correlation length of 500km between the sub-regions.

## 2.4 Sensitivity of Results to Changes in Assumed OH Distributions

To examine the sensitivity of our results to changes in the magnitude of OH, we run a single sensitivity run that is made up of two parts. First, we imposed a 0.5%/yr uniform decrease to our 3-D OH field from 2004 to 2019, consistent with the 7% reduction over 2003–2016 proposed by Turner et al. (2017). Second, we imposed a larger global-scale OH reduction of 1.5% in 2020 based on recent studies (Miyazaki et al., 2021; Laughner et al., 2021) to describe a more abrupt change due to widespread reductions in nitrogen oxides ($NO_x$) associated with closing down manufacturing during the first Covid-19 lockdown (Cooper et al., 2022). Newer studies have suggested the OH reduction in 2020 was closer to 1% Peng et al. (2022); Feng et al. (2023), but these estimates are also subject to uncertainties. The purpose of this numerical experiment is to determine the sensitivity of *a posteriori* $CH_4$ flux estimates to changes in assumed variations in OH and not to issue an proclamation about a time profile of OH that would simultaneously fit observed changes in $CH_4$ and $\delta^{13}$C-$CH_4$.

We use the Siegel linear non-parametric estimates (Siegel, 1980) to fit a line to our *a posteriori* $CH_4$ emissions from 2004 to 2020. This method is less sensitive to outliers, e.g. El Niño, that would otherwise compromise the linear trend estimate (Palmer et al., 2021), and the resulting linear trend estiate has lower variables that simpler methods. We find Siegel trend estimates are similar to those estimated by the Theil-Sen estimator.

## 3 Results

Here, we report *a posteriori* estimates for total $CH_4$ emissions inferred from *in situ* and GOSAT data and then the corresponding *a posteriori* isotopic source signatures for $\delta^{13}$C. We draw comparisons with previous studies throughout this section.

### *A posteriori* emission estimates of total $CH_4$

Figure 2 shows the annual mean differences in regions between *a priori* emission estimates and *a posteriori* emission estimates for both ground-based (2004–2020) and GOSAT results (2009–2020). Absolute emissions values are plotted in Figure 3 and shown in Table 2 for completeness.

On a global scale, terrestrial *a posteriori* emissions inferred *in situ* and GOSAT data have progressively increased relative to *a priori* values since about 2014. The peak difference is in 2020 when we find increased emissions relative to *a priori* emissions of $68.5 \pm 61.5$ Tg/yr in 2020 for the *in situ* inversion and $61.5 \pm 37.3$ Tg/yr higher emissions for the GOSAT inversion. Global ocean $CH_4$ emissions inferred *in situ* and GOSAT data support a negative bias in the *a priori*, which we do not discuss further.

As a zeroth order check of our *a posteriori* emission estimate of total $CH_4$, Figure 4 we compare the published NOAA atmospheric growth rate of $CH_4$ with our corresponding *a posteriori* atmospheric mole fractions. Generally, we find the *a posteriori* values inferred from *in situ* and GOSAT data are consistent with the overall trend of the changes in the growth rate, with large year-to-year changes that we explain now in terms of regional emission changes.

Changes in the global terrestrial emissions reflects changes from different geographical regions. Differences between *a posteriori* emission estimates inferred from *in situ* and GOSAT data are partly due to differences in the geographic coverage of

the datasets. Ground-based data have poorer geographic coverage, particularly over the tropics and the southern hemisphere and satellite data are currently available at most once per day in cloud-free conditions. Using the *in situ* data, we find that the largest *a posteriori* emission increases over the 2004 to 2020 period (Table 2), determined by the Siegel linear estimator, are over the tropics (3.6 Tg/yr/yr, comprising North Africa, southern Africa, tropical South America, and Tropical Asia), followed by China (1.6 Tg/yr/yr) then by small contributions (individually <0.2 Tg/yr) from elsewhere.

Table 1 provides an overview our annual mean sector-based *a posteriori* emissions for 2004–2020. Generally, our values are close to the reported median values and within the range of values reported by Saunois et al. (2020).

Over the tropics, there is broad consistency between GOSAT and *in situ* data (Figure 2) that highlights the negative bias in the *a priori* over Northern Africa (bias of -8.6 Tg/yr), Tropical Asia (bias of -7.2 Tg/yr), and Tropical South America (bias of -11.63 Tg/yr). The *in situ* and GOSAT data for China support a small, steady increase in emissions from 2009 to 2020 (1.0 Tg/yr), with emissions inferred from the GOSAT data generally smaller than *a priori* values throughout the period (Figure 2). Data over India have a small mean annual trend (0.33 Tg/yr). *In situ* and GOSAT data are more consistent in sign (but not magnitude) at temperate latitudes (Figure 2). *A posteriori* emissions from *in situ* and GOSAT data are generally lower by more than 12.0 Tg/yr and 5.6 Tg/yr, respectively, over Temperate North America and higher by more than 13.0 Tg/yr and 7.0 Tg/yr, respectively, over Temperate Eurasia, with the smallest discrepancies relative to the *a priori* before 2009. *A posteriori* emissions from boreal regions appear to be larger than *a priori* values by 4.2 Tg/yr before 2009 (Figure 2). After 2009, *in situ* data become progressively more consistent with the *a priori* over North America and is typically smaller than *a priori* values over Eurasia by $\simeq$2.6 Tg/yr. GOSAT appears to show the converse situation: after 2009, data are lower than *a priori* values by 4.4 Tg/yr over North America and comparable with *a priori* values over Eurasia. In the southern hemisphere, *in situ* data closely *a priori* values, as expected, since there are few places where data are collected. GOSAT data show a small but persistent increase in emissions with time over Southern Africa (0.41 Tg/yr), highlighting the negative bias in *a priori* emissions over Australia and over Temperature South America.

We use the *a posteriori* error covariance matrix from our MAP inversion ($\mathbf{A}$, equation 2) to determine our ability to independently estimate $CH_4$ emissions from our geographical regions. Figure A1 shows no significant *a posteriori* correlations between neighbouring geographical regions in our state vector. This is consistent with the *in situ* data being able to estimate independently regional emission estimates in our state vector.

Our *a posteriori* emission estimates are broadly consistent with previous studies. For example, the increase in tropical emissions has been reported using GOSAT data or *in situ* data within a 3-D CTM inversion (McNorton et al., 2016; Fujita et al., 2020), which examined shorter time periods of 2003–2015 and 1995–2013, respectively. The increase over Eastern Africa (that lies within our North Africa region) has been reported by several studies (Lunt et al., 2019, 2021; Pandey et al., 2021; Feng et al., 2022). Sheng et al. (2021) reported using GOSAT data that $CH_4$ emissions from China increased by 0.36 Tg/yr from 2012 to 2017. Over the same time period, we estimate an increase of 0.64 Tg/yr and 0.50 Tg/yr inferred from *in situ* and GOSAT data, respectively.

Figure A2 shows observed $CH_4$ timeseries at ground-based sites that we use to determine the corresponding GEOS-Chem *a priori* and *a posteriori* mole fractions. *A priori* values already show excellent agreement with observations (mean residual of

14.1 ppb and root-mean-square error (RMSE) of 18.3 ppb), but this is generally improved after the model is fitted to the *in situ* data, with smaller mean residuals (12.5 ppb) and RMSE (17.0 ppb). This is consistent previous studies such as McNorton et al. (2018) that reported *a posteriori* RMSE values of 12.3 ppb. Figure A3 shows *a posteriori* $CH_4$ mole fractions at NOAA sites that we do not include the inversion. This provides an additional and independent test of our ability to describe atmospheric $CH_4$ using a subset of NOAA data that we use in our inversion (Table A2). Generally, our *a posteriori* estimates agree with these independent data, but for some sites the model has difficulty reproducing the data, e.g. AMY (western S. Korea), KZD (Kazakhstan), and SDZ (mainland China). This is because some sites are influenced by local sources that are not representative of the spatial scale of our transport model ($\simeq$50,000 km$^2$) Similarly, we find agreement using *a posteriori* mole fractions using GOSAT data (Figure A4; mean residual of 29.1 ppb and RMSE 35.1 ppb).

### *A posteriori* source signatures of $\delta^{13}$C

Figure 5 shows *a posteriori* regional $\delta^{13}$C emissions source signatures inferred from ground-based *in situ* data. We group our results into approximately three-year bands, as a residual from the 2004–2007 mean value, to show how the regional isotopic source signatures change across the time series.

Relative to *a priori* emissions (Figure A5), *a posteriori* values from Northern Boreal regions (Boreal North America and Eurasia) have isotopically lighter signatures (-62‰), consistent with a larger contribution from isotopically lighter biogenic emissions and/or a smaller contribution from isotopically heavier thermogenic or pyrogenic emissions (Figure A5). Conversely, *a posteriori* values from regions such as Temperate Eurasia, Australia and Southern Africa have isotopically heavier source signatures (approximately -40‰), suggesting a larger proportion of thermogenic or pyrogenic emissions and/or a smaller contribution from isotopically light biogenic emissions.

Figure 5 shows a general trend towards isotopically lighter regional source signatures of $\delta^{13}$C across the time series. Our analysis suggests this trend has been ongoing since 2012 and is observed in all regions worldwide, but is strongest (compared with *a priori* estimates) over Tropical and Southern Hemispheric regions. For example, Tropical South America and Southern Africa are 1.2‰ and 0.9‰ isotopically lighter than *a priori* values for for 2019 and 2020, respectively.

Our analysis also highlights a period 2007-2012 when regional source signatures, particularly northern hemisphere regions, become isotopically heavier compared with *a priori* source signatures (by 1.0‰ 2007–2009; by 0.8‰ 2010–2012; and 0.3‰ 2013–2015). After 2012, regional source signatures of $\delta^{13}$C generally become isotopically lighter. This result is suggestive that 2012 was period when there was a change in the balance of global sources that determine changes in atmospheric $CH_4$. These isotopic shifts in 2008 and 2012 are noted by Nisbet et al. (2016), who used a box model and examine data from sites measured by NOAA and Royal Holloway, University of London (RHUL). They found that changes in removal rates could not explain these anomalies so that these events were attributed to changing emissions. We find that China experiences a weaker shift in 2012 to a ( 0.1‰) isotopically lighter $\delta^{13}$C source signature compared to *a priori* values (Figure 5) and compared to other temperate regions. This suggests that heavier isotopic source signatures (such as coal mines) play a larger contribution to this region.

Unlike the *a posteriori* total $CH_4$ emission estimates, we find significant *a posteriori* correlations between neighbouring regions for $\delta^{13}C$ source signatures (Figure A6). For example, there is a correlation of 0.95 between estimates for Southern Africa and Temperate South America so these cannot be considered as independent estimates. This result aligns with Basu et al. (2022) who used $CH_4$ mole fraction and $\delta^{13}C$ measurements to determine that tropical biogenic sources are driving $CH_4$ growth. They acknowledged that measurement coverage limited conclusions based exclusively on isotope ratio measurements. Nevertheless, they found a clear trend of stronger emissions of isotopically lighter $CH_4$, indicative of an increased role for biogenic emissions in the global source makeup.

We find that *a posteriori* regional $\delta^{13}C$ source signatures result in a time series of $\delta^{13}C$ that is more consistent with observations than *a priori* values (Figure A7), as expected. This particular affects the period 2008–2018 when *a priori* emissions source signatures are significantly isotopically lighter. Our *a posteriori* source signatures result in a mean observed-model residual and RMSE of 0.11‰ and 0.14‰, respectively. These are smaller than those corresponding to *a priori* values for the observed-model residual (0.37‰) and RMSE (0.41‰). Our comparison is consistent with McNorton et al. (2018) (RMSE 0.1‰) and Fujita et al. (2020) (RMSE 0.08-0.25‰).

**Sensitivity to assumptions about OH**

Figure 6 shows the result from our sensitivity test that assumes 0.5%/yr uniform decrease to our 3-D OH field from 2004 to 2019, followed by a more abrupt decrease of -1.5% in 2020 to describe the widespread reduction in nitrogen oxide emissions. This is an idealised sensitivity test that is inconsistent with global-scale constraints on estimates of the global mean atmospheric growth of atmospheric $CH_4$, i.e., most of the observed global growth in atmospheric $CH_4$ can be explained by the changes in OH. Nevertheless this test provides us with some idea of the robustness of our results against changes in OH.

We find that this alternative assumption about OH does not significantly affect our results until much later in the timeseries (2017–2019), reflecting our large *a posteriori* uncertainties. We find a similar quality of fit of the *a posteriori* model to the data with or without considering the OH trend (not shown). This does not preclude a role for changes in OH but the concurrent *a posteriori* shifts in $CH_4$ emissions and regional isotopic source signatures of $\delta^{13}C$ are consistent with decreasing OH playing a smaller role than increasing emissions with isotopically light $\delta^{13}C$ source signatures in determining observed changes in atmospheric $CH_4$ (Lan et al., 2021).

The larger, abrupt change in 2020 results in a marked reduction (approximately 6%, 40 Tg/yr) in the emissions necessary to explain the increase in atmospheric $CH_4$. There is still debate about the impact of *a posteriori* $CH_4$ methane emissions. Peng et al. (2022) used *in situ* data and concluded that the increase in atmospheric $CH_4$ in 2020 could be attributed approximately equally between a decrease in OH and an increase in OH. Analysis of GOSAT suggest that increased emissions play a larger role (Qu et al., 2022; Feng et al., 2023).

## 4 Conclusions

We estimated regional $CH_4$ emissions and $\delta^{13}C$ source signatures for the period 2004–2020, inclusively, by fitting the GEOS-Chem 3-D atmospheric chemistry transport model to surface mole fraction data (Figure 1) and GOSAT atmospheric column data (2010-2020) using Bayesian inverse methods. We used surface sites for which we had complete monthly coverage over most of the study period (Table A2). Collectively, our results indicate that the post-2007 increases in $CH_4$ emissions are best explained by a progressive latitudinal shift in emissions from the northern midlatitudes to tropical latitudes. *A posteriori* $CH_4$ emission estimates inferred from the ground-based and GOSAT data show larger tropical emissions, particularly over North Africa, Tropical Asia, and Tropical South America, and over China and at the same time as mid-latitudinal emission proportions decreases. Source signature estimates inferred from the $\delta^{13}C$ measurements (Figure 1) over the same time period indicate that the latitudinal shift in $CH_4$ emissions is due to larger proportion of sources with a lighter atmospheric $\delta^{13}C$ signature (e.g., biogenic source such as wetlands) and/or a smaller proportion of sources with a heavier atmospheric $\delta^{13}C$ signature (e.g, thermogenic or pyrogenic sources). Our results are broadly consistent with previous studies that focus on shorter, contributing periods (McNorton et al., 2018; Nisbet et al., 2019; Fujita et al., 2020; Yin et al., 2021; Lan et al., 2021; Basu et al., 2022)), providing confidence in our model assumptions and data selection. We find that our main results are robust against assuming a 0.5%/yr OH decrease from 2004 to 2019, consistent with Turner et al. (2017), followed by an abrupt 1.5% OH drop in 2020 that reflects the widespread decrease in nitrogen oxide emissions from shutting down manufacturing during the first Covid-19 lockdown. This is an idealised sensitivity test but nevertheless provides us with some idea of the robustness of our results against changes in OH. A more detailed discussion of the role of OH in 2020 is discussed elsewhere (Qu et al., 2022; Peng et al., 2022; Feng et al., 2023).

Sparse geographic coverage of ground-based data results in larger uncertainties for regional emission estimates that are informed by fewer data, i.e. high and low latitudes in both hemispheres. For $CH_4$, this deficiency can be partly addressed using the satellite data, but isotope ratios cannot usefully be retrieved from Earth observation satellite instruments. We use only three measurement sites for $\delta^{13}C$ in the Southern Hemisphere, which have a continuous record over the period of study. A consequence of this data sparcity is strong correlations between source signatures from neighbouring regions (Figure A6). We assume mean sectoral $\delta^{13}C$ source signatures from Sherwood et al. (2017). These values are highly uncertain, as different sectors produce a range of possible $\delta^{13}C$ values, and there are significant overlaps between recorded source signatures (Douglas et al., 2017), but the values chosen represent our current best knowledge of mean values. These data have greater value when they are used in a broader context with other data, as we have described in this study. We have used satellite observations to help identify that large-scale emission changes over regions that coincide with wetlands.

Collectively, empirical evidence, including *in situ* and GOSAT observation of $CH_4$ and *in situ* $\delta^{13}C$ data, points to an increasing biogenic source originating from the tropics. While we cannot definitively attribute these changes to increasing wetland emissions, there is sufficient contextual evidence, building on previous studies, to suggest that wetlands are playing a significant role in recent growth of atmospheric $CH_4$. First, large changes in OH that would needed to explain this atmospheric growth are inconsistent with increasingly isotopically light $\delta^{13}C$ observations in the atmospheric record (Lan et al., 2021).

Second, we know from *in situ* data the broad geographical regions responsible for increasing $CH_4$ emissions and isotopically lighter $\delta^{13}C$ source signature, where the seasonal cycles are consistent with biogenic emissions peaking outside the burning season. Third, GOSAT provide us with additional information about the geographical distribution of $CH_4$ emissions: tropical emission hotspots are colocated with known wetland regions (Lunt et al., 2019, 2021; Pandey et al., 2021; Wilson et al., 2021; Feng et al., 2022, 2023; Hardy et al., 2023). Finally, we also have evidence from other satellite data, e.g., hydrology, that help explain the growth of wetland emissions in the last decade (Lunt et al., 2019; Feng et al., 2022). Greater confidence in source attribution of changes in atmospheric $CH_4$ may come from collecting and interpreting $\delta D$ and multiply-substituted 'clumped' isotopes (Douglas et al., 2017; Chung and Arnold, 2021), alongside $\delta^{13}C$. This needs to be accompanied by field measurements of these isotope ratios to improve delineation between different sectors.

The evidence presented here is consistent with a growing body of work that points to a substantial increase in biogenic $CH_4$ emissions from the tropics. This increase will likely have major implications for our achieving the goals of the Paris Agreement (Nisbet et al., 2019). Nature does not care about the origin of atmospheric $CH_4$ so that increasing biogenic emissions will require larger emission reductions from anthropogenic sectors, placing additional pressure on citizens to reduce their carbon footprints.

## 5   Code and data availability

The community-led GEOS-Chem model of atmospheric chemistry and model is maintained centrally by Harvard University (http://geos-chem.seas.harvard.edu), and is available on request. The ensemble Kalman filter code is publicly available as PyOSSE (https://www.nceo.ac.uk/data-tools/atmospheric-tools/).

## 6   Data availability

All the data and materials used in this study are freely available. The NOAA-GML and CU-INSTAAR ground-based $CH_4$ and $\delta^{13}C$ data are available from the NOAA GML FTP server (https://gml.noaa.gov/dv/data), subject to their fair use policies. Data from JR-STATION network was provided with cooperation of NIES Japan. The University of Leicester GOSAT Proxy v9.0 XCH4 data are available from the Centre for Environmental Data Analysis data repository at (https://doi.org/10.5285/18ef8247f52a4cb6a14013f8235cc1eb), and from the Copernicus Climate Data Store. EDGAR data is available at (https://edgar.jrc.ec.europa.eu/), GFED-4 data is available at (https://www.globalfiredata.org/data.html), WETCHARTS data is available at (https://daac.ornl.gov/cgi-bin/dsviewer.pl?ds_id=1502).

## Appendix A: Isotopologue Emissions

To simulate the atmospheric isotope ratio $\delta^{13}C$ the isotopologues $^{12}CH_4$ and $^{13}CH_4$ are considered separately in the model. To calculate the specific sectoral isotopologue emissions we use the emissions calculated from the mole fraction $CH_4$ simulation and the isotope ratios defined in Table 1. We consider the isotope $^{13}C$ relative to all isotopes in the sample (designated thereafter

as $13x$) using:

$$13x = \frac{^{13}C}{^{12}C + ^{13}C} = \frac{^{13}C/^{12}C}{1 + (^{13}C/^{12}C)}, \tag{A1}$$

where $^{13}C/^{12}C$ is calculated from the $\delta^{13}$C reported on the international carbon isotope scale VPDB (Vienna Pee Dee Belemnite). This is the proportional molar abundance of the isotopologues containing $^{13}C$ (dominated by $^{13}$CH$_4$) relative to the isotopologues containing $^{12}C$ (dominated by $^{12}$CH$_4$) . This value has to be adjusted before being applied in GEOS-Chem to convert from isotope ratio values to kg values used by emission inventories:

$$SF13 = 13x \times \frac{M_{13}}{M_{tot}}, \tag{A2}$$

where $SF13$ is the scale factor applied to each emissions type for the $^{13}$CH$_4$ simulation, $M_{13}$ is the molecular weight of $^{13}$CH$_4$ (17.035 g/mol) and $M_{tot}$ is the molecular weight of CH$_4$ (16.04 g/mol).

For the $^{12}$CH$_4$ counterpart to $^{13}$CH$_4$, we use a similar approach. The ratio of $^{12}C$ compared with all isotopes in the sample (designated as $12x$) is given by:

$$12x = \frac{^{12}C}{^{13}C + ^{12}C}. \tag{A3}$$

This is similarly adjusted from molar to mass ratio; $SF12$ is the scale factor for each emissions type in the $^{12}$CH$_4$ simulations:

$$SF12 = 12x \times \frac{M_{12}}{M_{tot}}, \tag{A4}$$

where $M_{12}$ is the molecular weight of $^{12}$CH$_4$ (16.03 g/mol). Since $^{13}C$ and $^{12}C$ are the only stable carbon isotopes of CH$_4$, $13x$ and $12x$ should sum to 1.

*Author contributions.* A.D. led the data analysis with contributions from P.I.P. and L.F. A.D. and P.I.P. led the writing of the paper with contributions from L.F. and T.A.. X.L., S.M., R.P. and H.M. provided data.

*Competing interests.* The authors declare that they have no competing interests.

*Acknowledgements.* A.D. is supported by the University of Edinburgh's E3 Doctoral Training Partnership, funded by the Natural Environment Research Council, and by a contribution from the National Physical Laboratory (NPL). P.I.P., L.F and R.P. acknowledge support from the UK National Centre for Earth Observation funded by the Natural Environment Research Council (NE/R016518/1 and NE/N018079/1) and the Copernicus Climate Change Service (C3S2_312a_Lot2). TA acknowledges support from the Natural Environment Research Council

(NE/V007149/1) and the NPL Director's Fund. We thank NOAA ESRL and CU-INSTAAR for providing $CH_4$ and $\delta^{13}C$ data. We thank the Japanese National Institute for Environmental Studies and the Ministry of Environment for the GOSAT data and their continuous support as part of the Joint Research Agreements at the Universities of Edinburgh and Leicester. We also thank the GEOS-Chem community, particularly the team at Harvard who help maintain the GEOS-Chem model, and the NASA Global Modeling and Assimilation Office (GMAO) who provide the MERRA-2 data product.

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

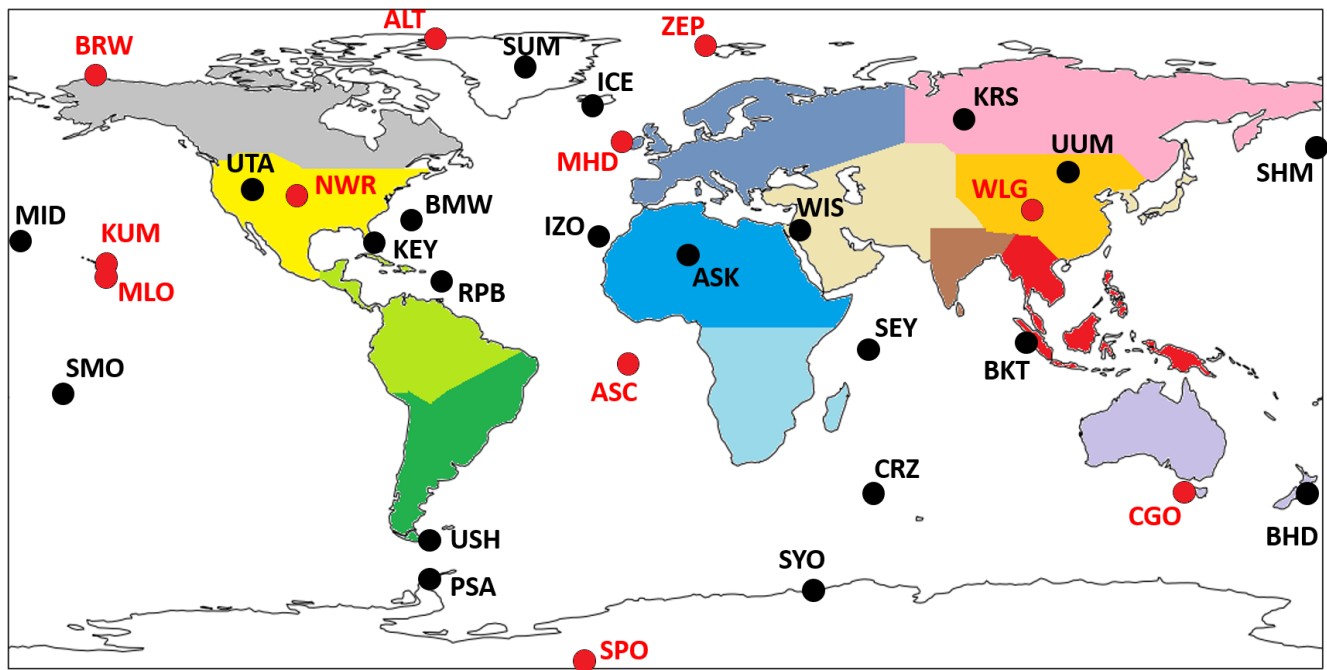

**Figure 1.** Map showing regions that are optimised in the $CH_4$ and $\delta^{13}C$ inversions, in different colours. Black dots and labels show the location of ground-based measurement sites that measure $CH_4$ mole fraction. Red dots and labels indicate both mole fraction $CH_4$ and $\delta^{13}C$ measuring sites. Regions are named as follows: Grey - North American Boreal; Yellow - North American Temperate; Light Green - South American Tropical; Dark Green - South American Temperate; Purple - Europe; Blue - North Africa; Light Blue - Southern Africa; Pink - Boreal Eurasia; Orange - China; Brown - India; Peach - Temperate Eurasia; Red - Tropical SE Asia; Lilac - Oceania; White - Oceans. Site identifiers are detailed in Table A2.

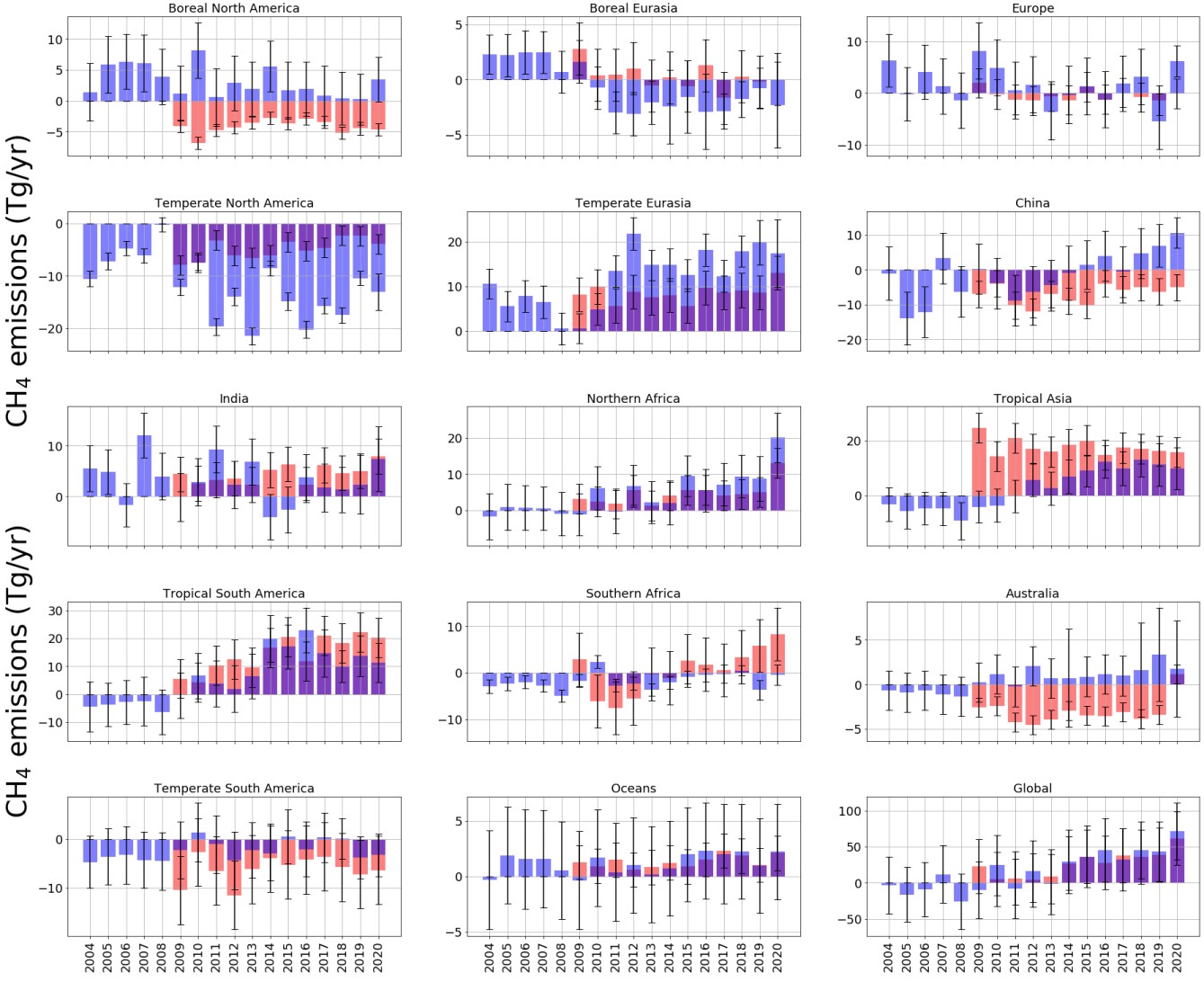

**Figure 2.** Annual mean CH₄ *a posteriori* emissions estimates as a residual value relative to *a priori* (Tg/yr) from each of the inversion regions in latitudinal order (geographic coverage indicated by Figure 1), for both ground-based and GOSAT inversion results. Uncertainties are indicated, as calculated from inversion calculations, with a *a priori* uncertainty of 50% for the ground-based results and 60% for the GOSAT results. The ground-based *a posteriori* is in blue; the GOSAT *a posteriori* are in red.

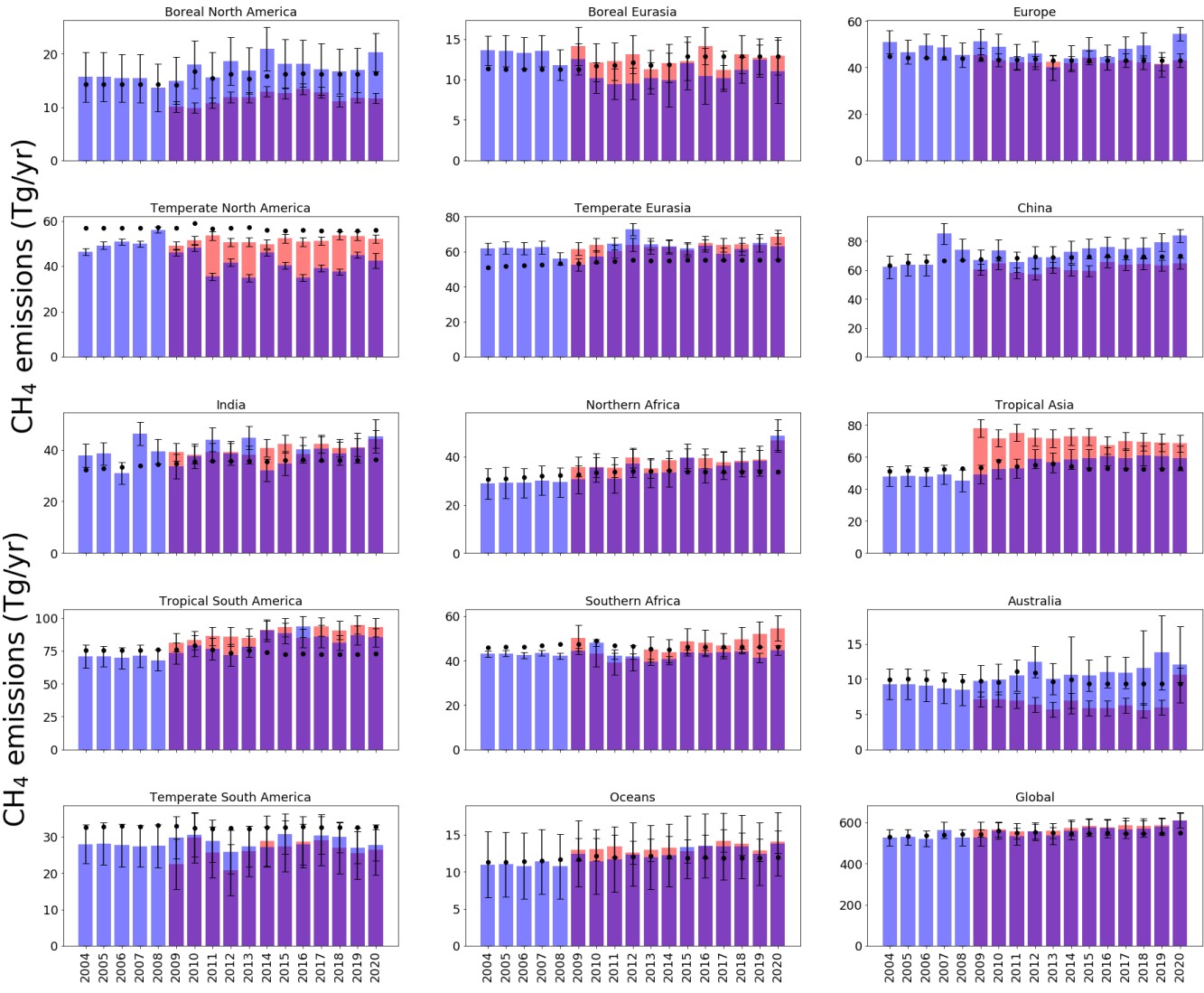

**Figure 3.** *A posteriori* emissions estimates (Tg/yr) inferred from ground-based *in situ* data (blue) and GOSAT data (red, with record starting in 2010) for the geographical regions shown by Figure 1. *A priori* emissions estimates are denoted by black dots and *a posteriori* uncertainties are denoted by whisker bars.

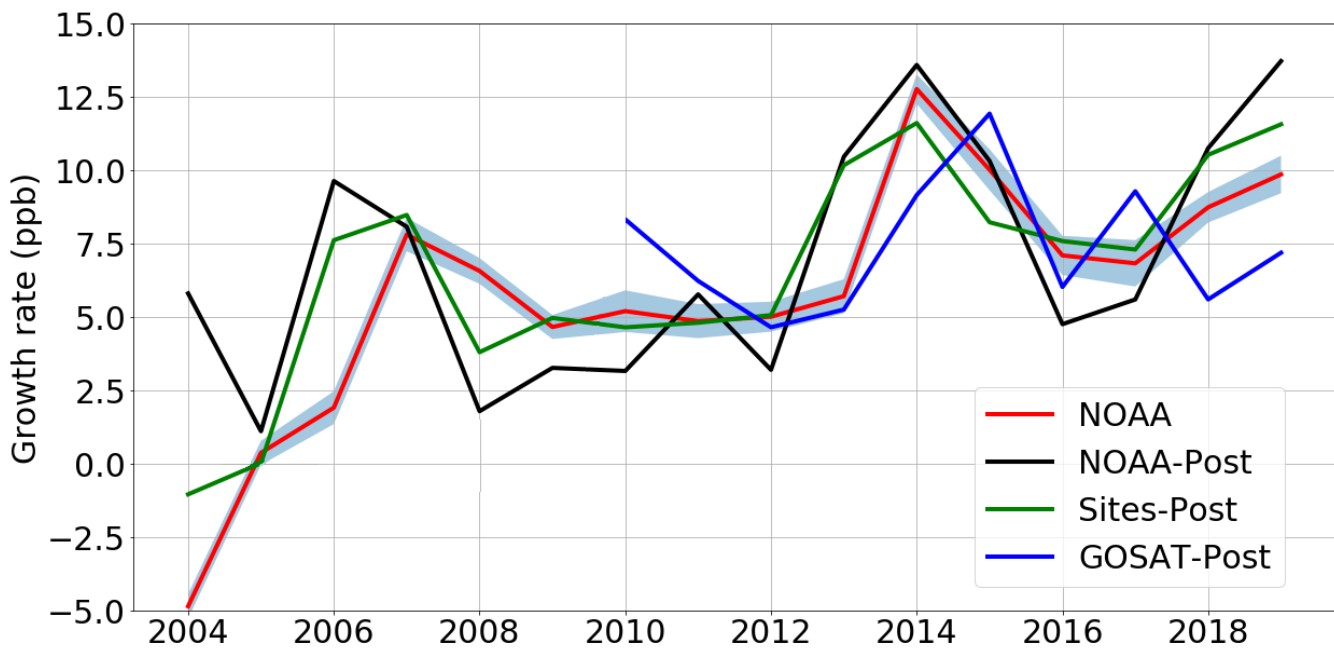

**Figure 4.** *A posteriori* annual mean atmospheric $CH_4$ growth rate inferred from *in situ* (black line) and GOSAT data (blue line) compared with the equivalent data as published by NOAA (red line, with uncertainty as blue surrounding field, Dlugokencky et al., 2020). The green line denotes the annual atmospheric growth rate determined using the *in situ* mole fraction data from the sites included in the inversion ('Sites-Post'). To calculate the atmospheric growth rates from model calculations (Ground-Post and GOSAT-post), we compare the average global $CH_4$ mole fraction in one year (the mean mole fraction of every grid box in every month of a year), with the mean value from the following year. The calculation is January-January, in order to remove the effects of the seasonal cycle, following the approach by NOAA (Dlugokencky et al., 2020).

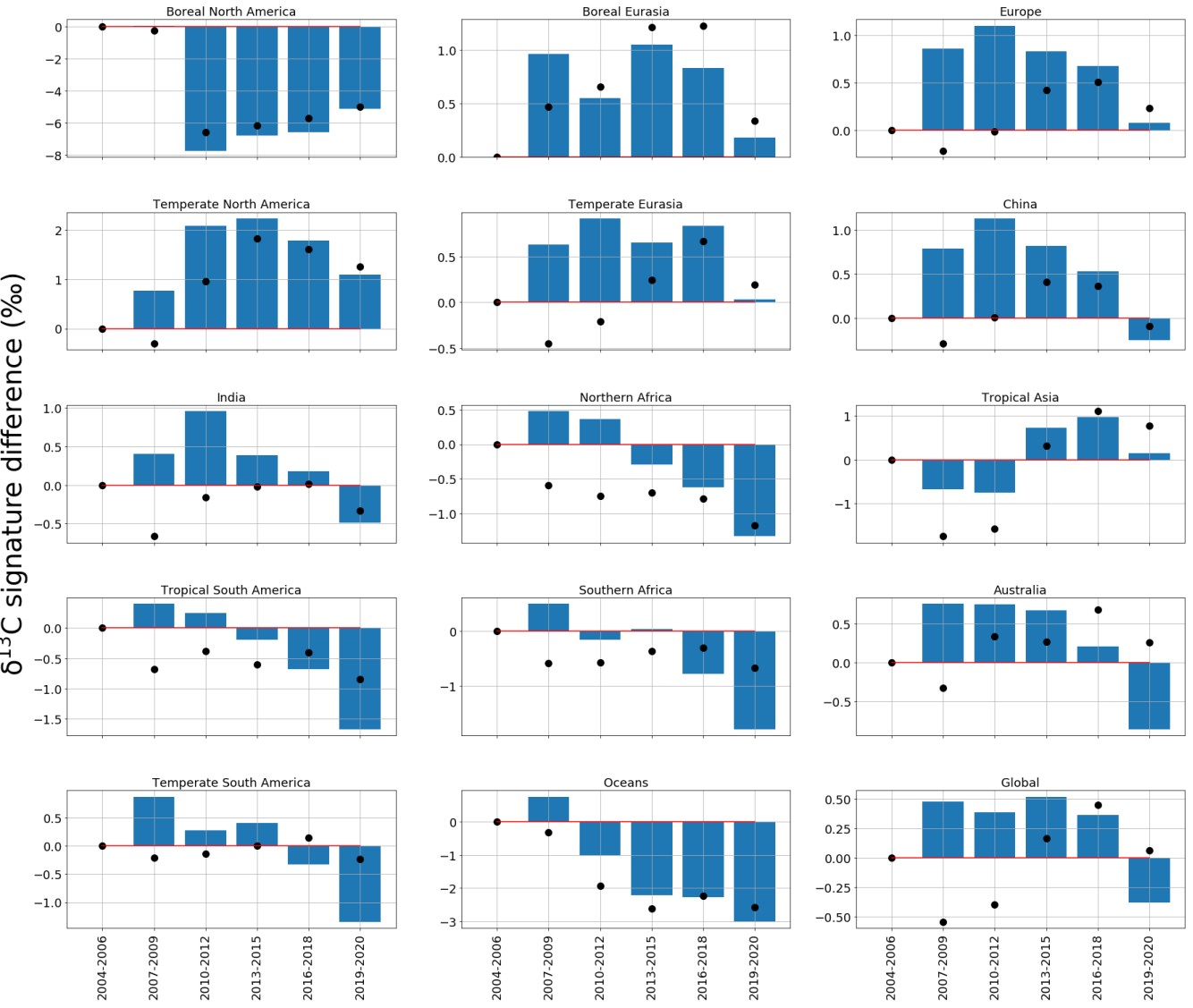

**Figure 5.** Regional and global *a posteriori* $\delta^{13}$C emissions source signatures (‰), in three-yearly groups (2004-06, 2007-09, 2010-12, 2013-15, 2016-18, 2019-20) as a residual from the 2004-06 *a posteriori* regional emissions source signature value. The *a priori* equivalent is represented by black dots. The regions are those solved for in the CH$_4$ and $\delta^{13}$C inversions and are indicated by Figure 1.

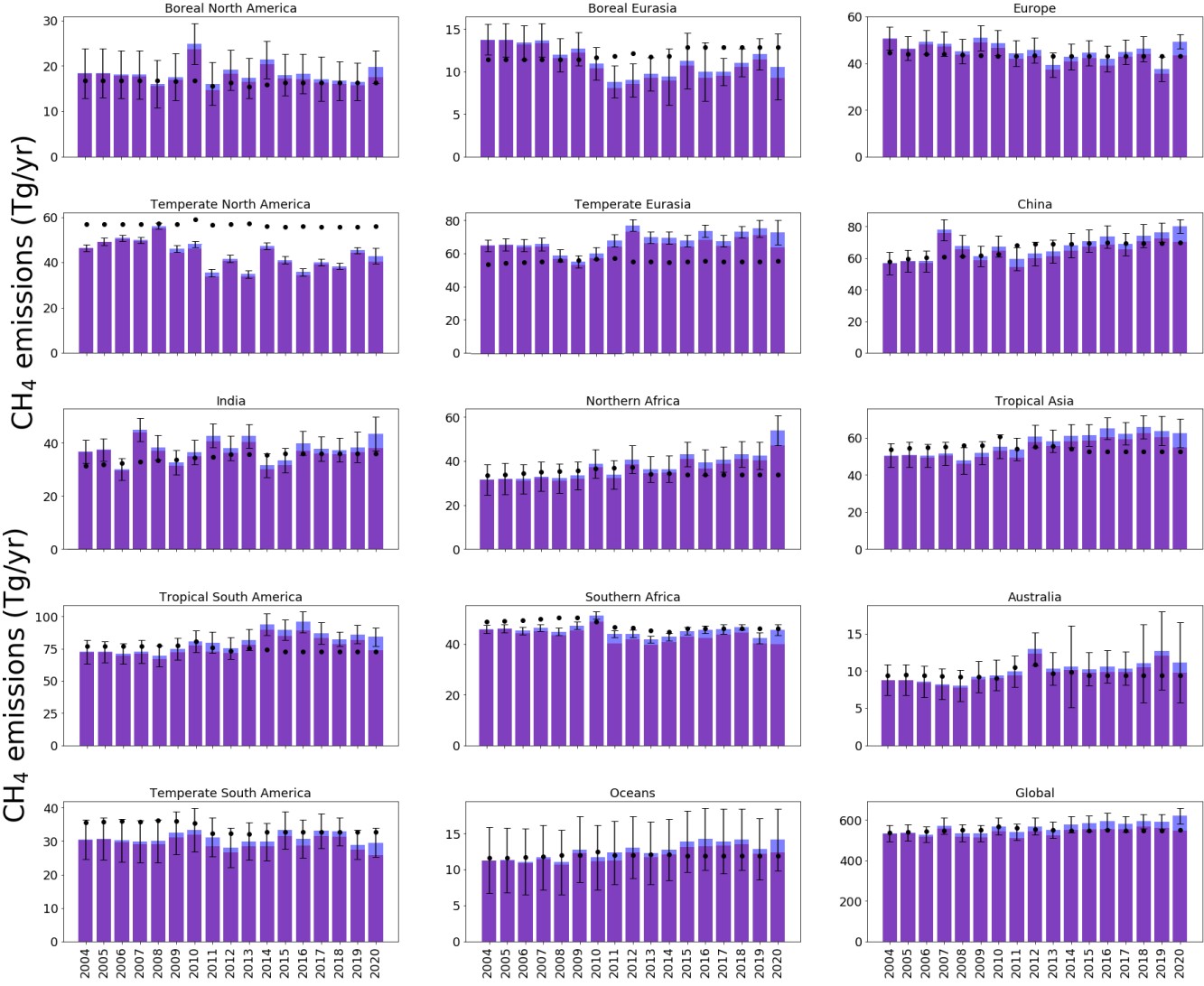

**Figure 6.** Annual mean CH₄ emissions (Tg/yr) for each region of the inversion (indicated by Figure 1) inferred from the ground-based data (dark blue) and the emissions estimates determined by a reduced OH values (described in the text, shown in red). *A priori* regional emissions estimates are indicated by black dots. Regional uncertainties for the *a posteriori* emissions are indicated.

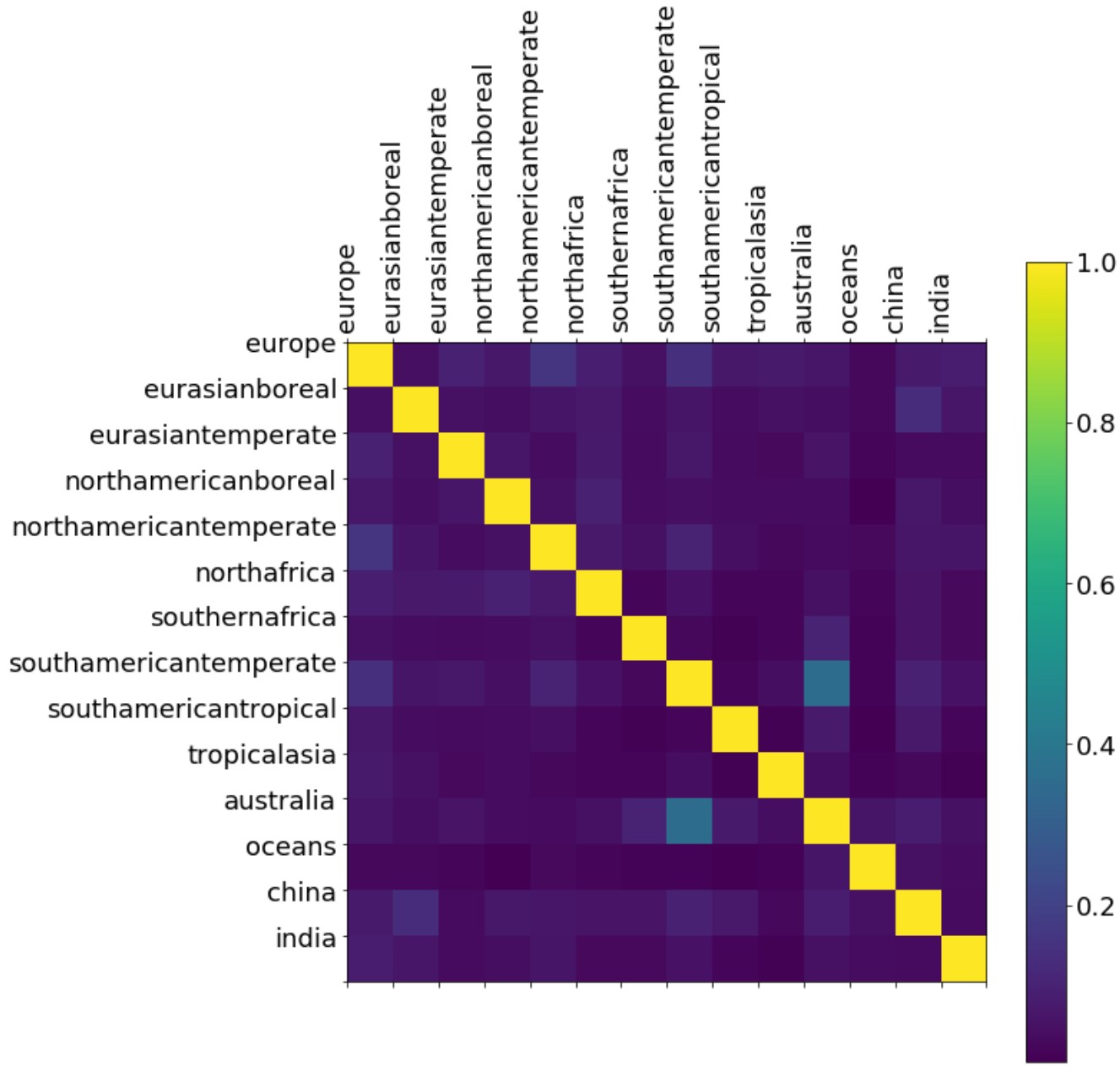

**Figure A1.** *A posteriori* correlations between CH$_4$ emissions from geographical regions inferred from ground-based CH$_4$ mole fraction data. These correlations are determined by normalising the diagonal elements of the *a posteriori* error covariance matrix (Eq. 2).

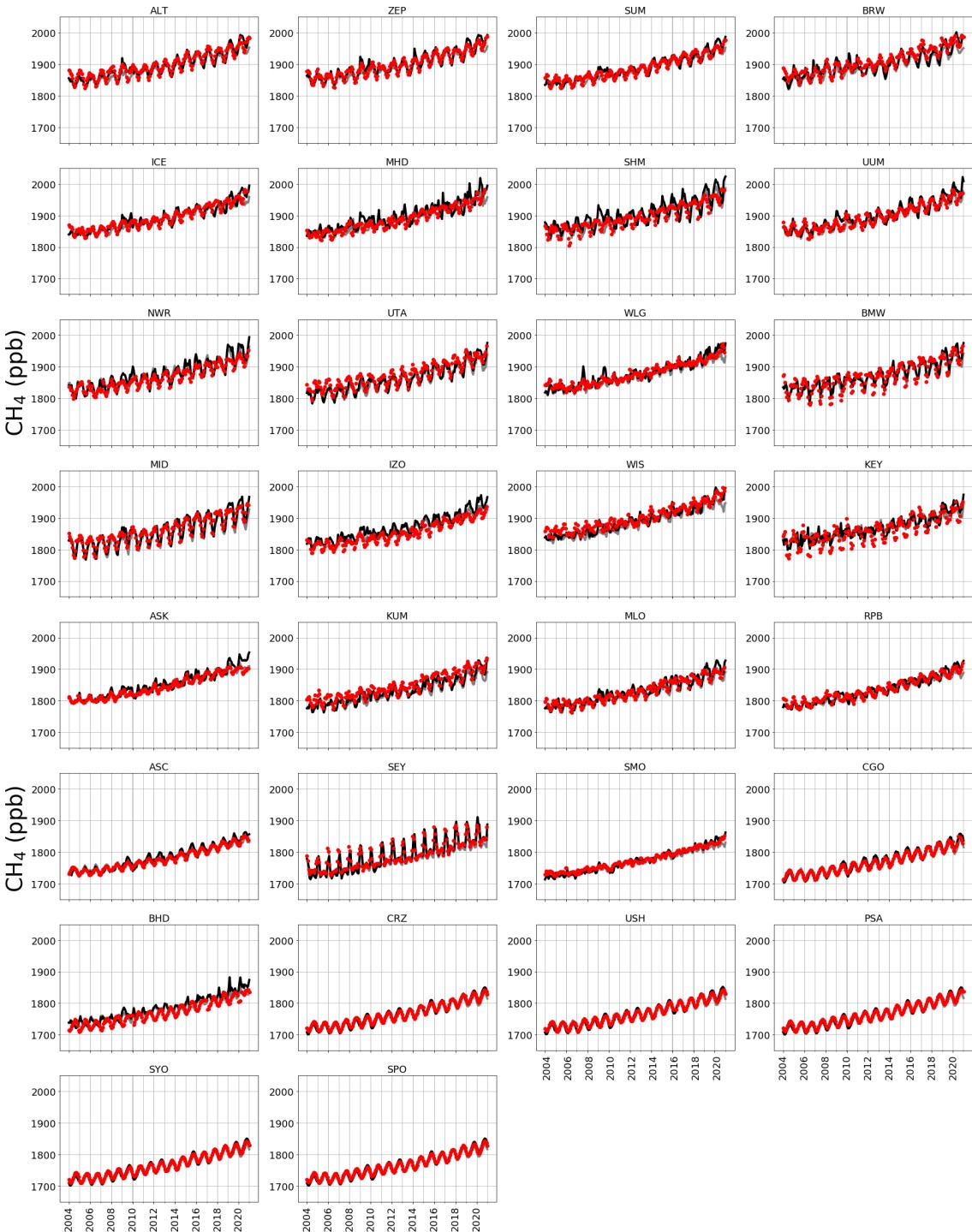

**Figure A2.** Observed (red dots), and *a priori* (grey), *a posteriori* (black) model atmospheric mole fractions at a series of NOAA sites (subplot titles denote site codes, Table A2), covering a range of latitudes.

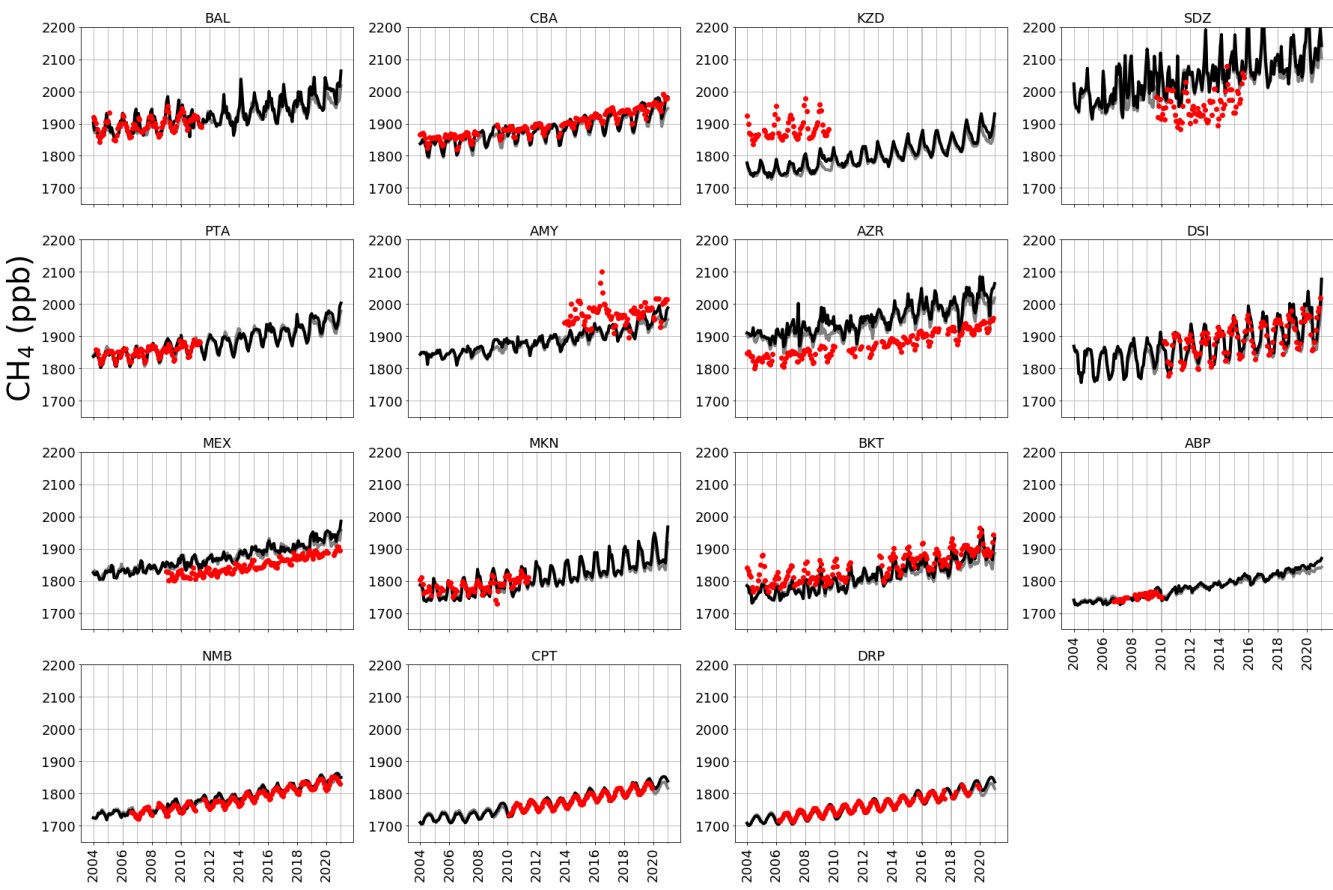

**Figure A3.** *A posteriori* (black) monthly estimates of atmospheric CH$_4$, simulated at NOAA sites across latitudes. Red dots indicate monthly mean CH$_4$ data from the NOAA network sites indicated. These sites were not included in the CH$_4$ inversion, but are shown here to provide independent validation of *a posteriori* emissions. The sites included are: Baltic Sea, Poland (55.35°N, 17.22°E); Cold Bay, Alaska (55.21°N, 162.72°W); Sary Taukum, Kazahkstan (44.08°N, 76.87°E); Shangdianzi, China (44.65°N, 117.12°E); Point Arena, USA (38.95°N, 123.74°W); Anmyeon-do, Republic of Korea (36.54°N, 126.38°E); Terceira Island, Azores (38.77°N, 27.37°W); Dongsha Island, Taiwan (20.70°N, 116.73°E); High Altitude Global Climate Observation Center, Mexico (18.98°N, 97.31°W); Mt Kenya, Kenya (0.06°S, 37.29°E); Bukit Kototabang, Indonesia (0.20°S, 100.31°E); Arembepe, Brazil (12.77°S, 38.17°W); Gobabeb, Namibia (23.58°S, 15.03°E); Cape Point, South Africa (34.35°S, 18.49°E); and Drake Passage (59.00°S, 64.69°W).

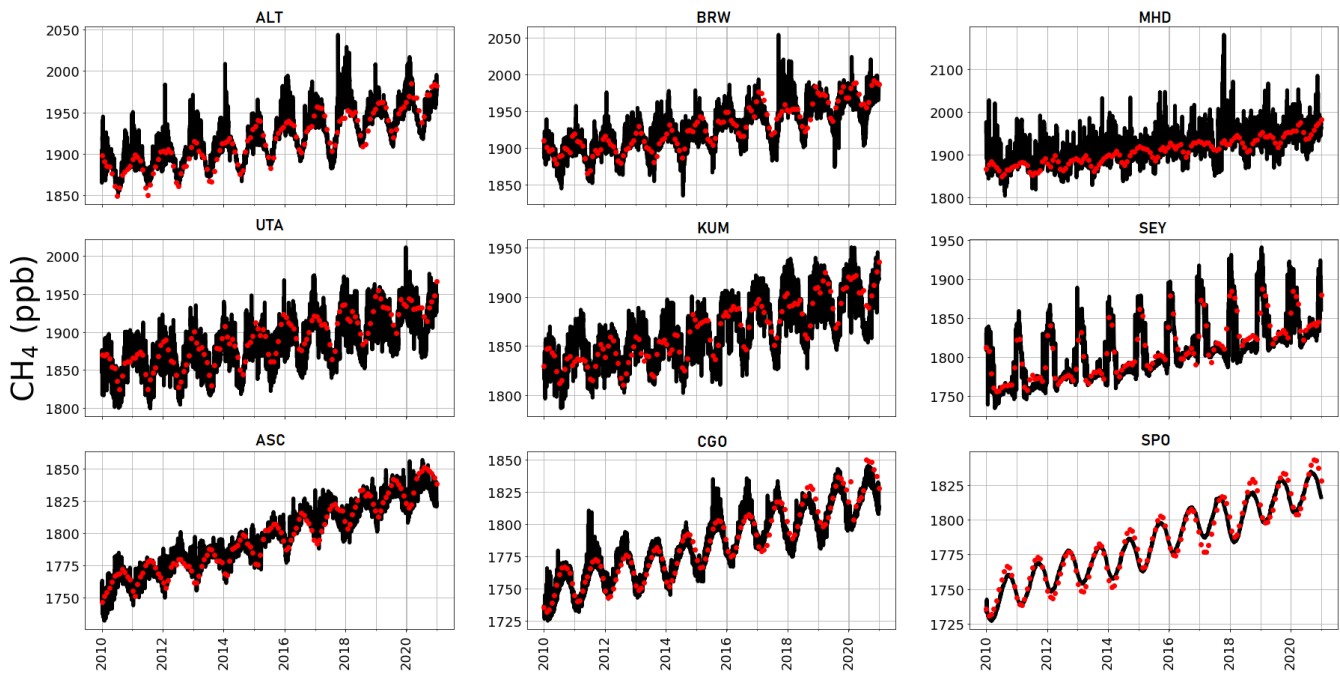

**Figure A4.** Observed (red dots), and three-hourly surface *a posteriori* CH$_4$ values inferred from GOSAT data (black) at the location of a number of NOAA sites (Table A2) 2010-2020.

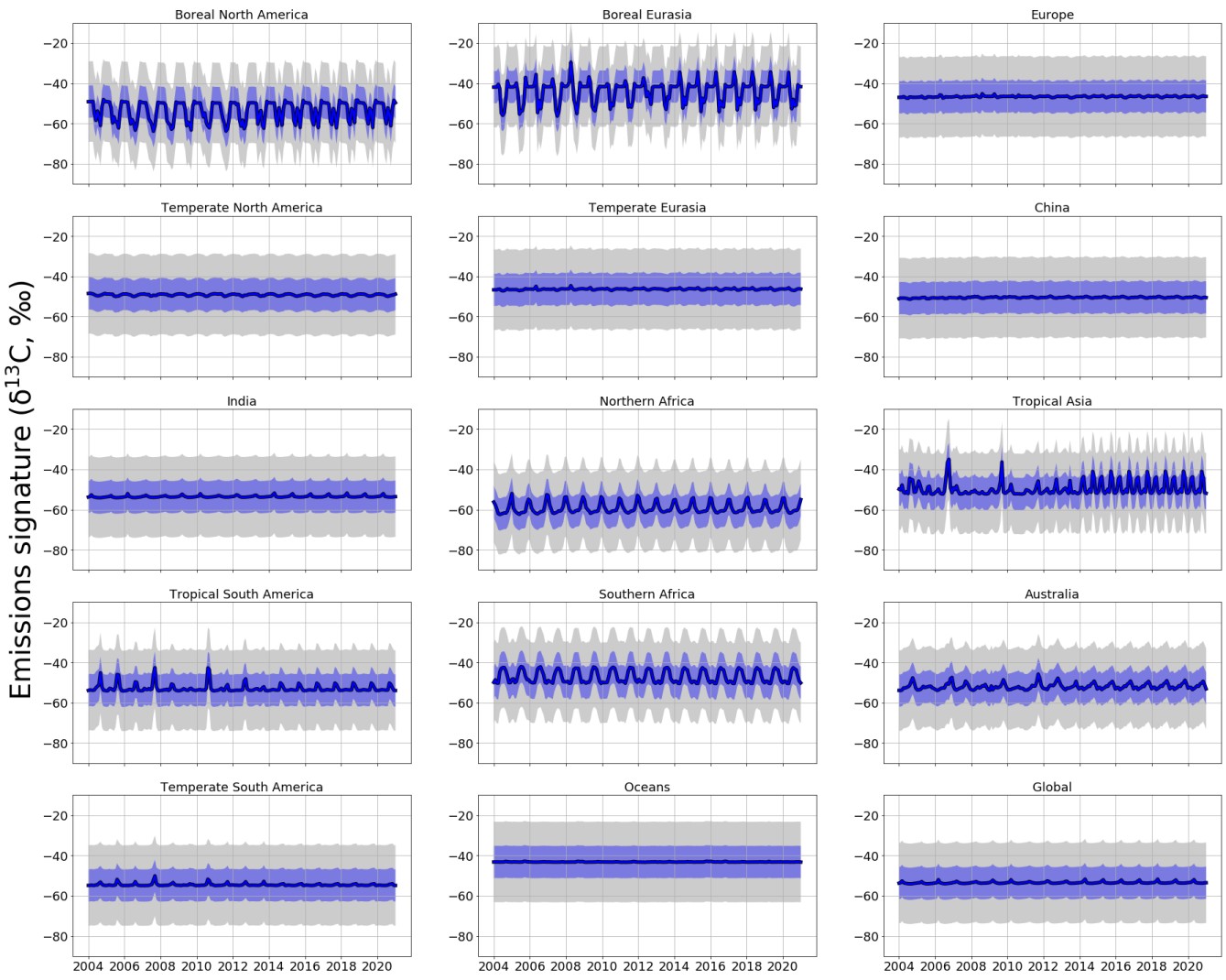

**Figure A5.** Monthly *a priori* (grey) and *a posteriori* (blue) regional $\delta^{13}C$ source signatures (‰). Values are produced using ground-based *in situ* $\delta^{13}C$ data. Uncertainties in source signatures are indicated as shaded envelopes, with *a priori* uncertainties of 15 ‰.

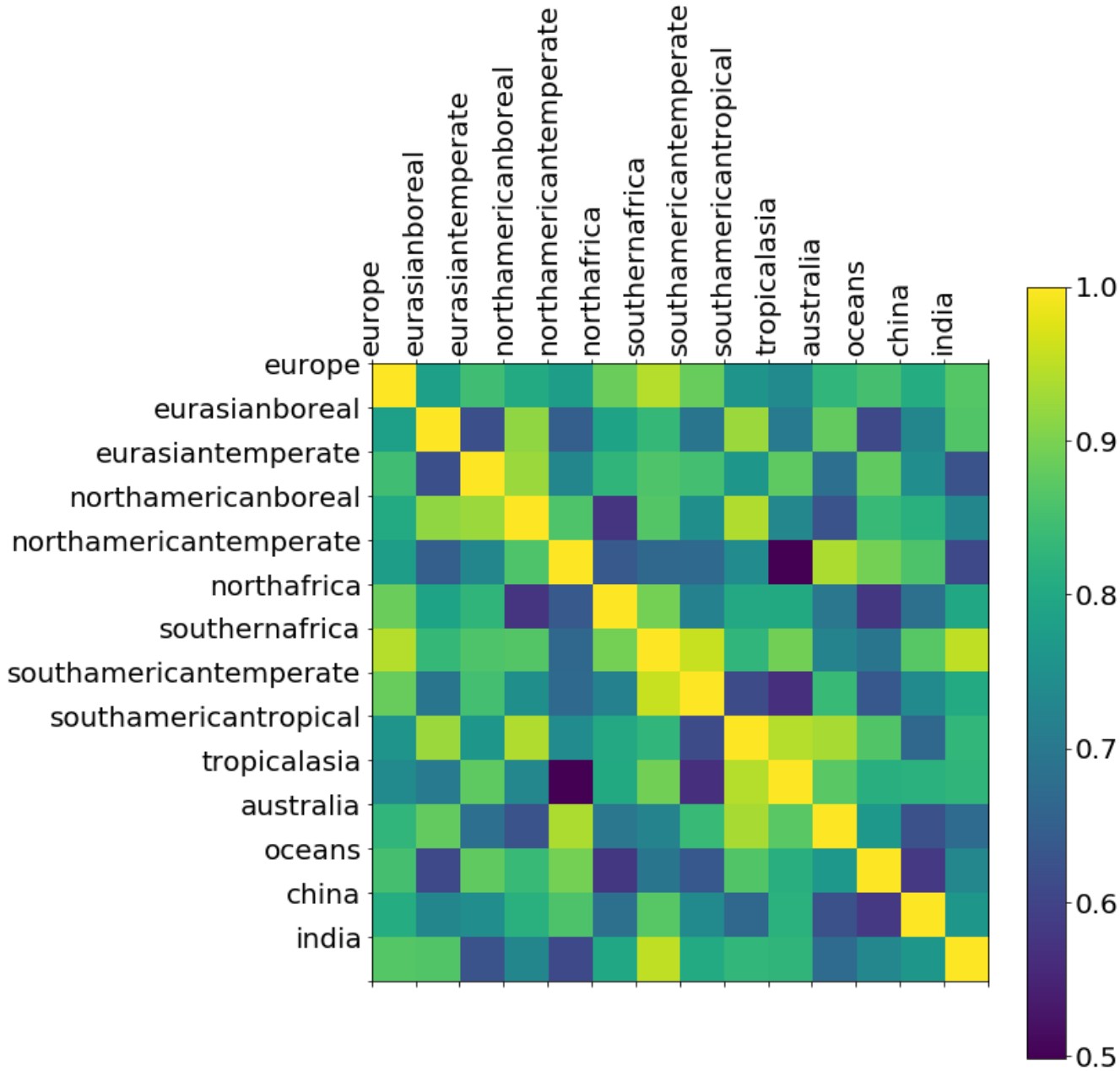

**Figure A6.** *A posteriori* correlations between $\delta^{13}$C source signatures from geographical regions inferred from ground-based $\delta^{13}$C data. These correlations are determined by normalising the diagonal elements of the *a posteriori* error covariance matrix (Eq. 2).

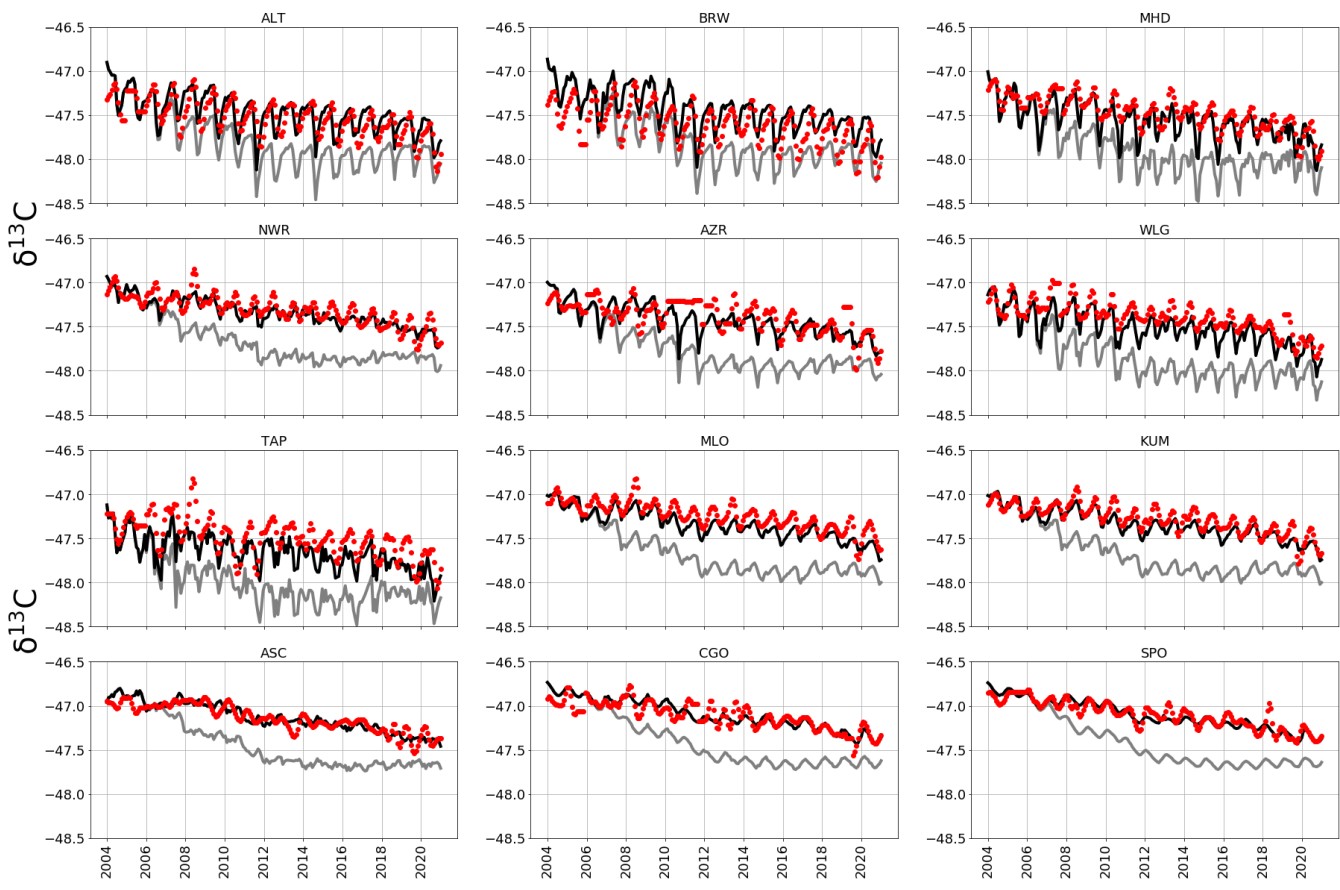

**Figure A7.** *A priori* (grey) and *a posteriori* (black) monthly estimates of atmospheric $\delta^{13}$C, simulated at NOAA sites across latitudes (site codes listed in Table A2). Red dots indicate monthly mean $\delta^{13}$C data from CU-INSTAAR for the respective sites.

**Table 1.** Global mean emissions of different CH$_4$ source types from bottom-up inventories (Saunois et al., 2020) and our *a posteriori* emission estimates, and the corresponding conventional isotope ratios signatures (Sherwood et al., 2017). Uncertainties are shown as max-min values in square brackets.

| Source Type | Annual Mean Emission (Saunois) 2003-2012 (Tg/CH$_4$) | Annual Mean Emission (This Study) 2004-2020 (Tg/CH$_4$) | Isotopic Ratio $\delta^{13}$C (‰) |
|---|---|---|---|
| Gas and Oil | 80 [68-92] | 82.0 | -44.0 [± 10.7] |
| Coal | 42 [29-61] | 53.7 | -49.5 [± 11.2] |
| Livestock | 111 [106-116] | 115.2 | -65.4 [± 6.7] |
| Waste | 65 [60-69] | 67.9 | -56.0 [± 7.6] |
| Biomass Burning | 17 [14-26] | 14.3 | -26.2 [± 4.8] |
| Termites | 9 [3-15] | 11.9 | -63.4 [± 6.4] |
| Wetlands | 149 [102-182] | 170.9 | -61.5 [± 5.4] (Tropical) -71.5 [± 5.4] (Arctic) |
| Rice | 30 [25-38] | 30.7 | -62.2 [± 3.9] |

**Table 2.** Annual regional mean *a posteriori* CH$_4$ emissions (Tg/yr) from 2004 to 2020. Regions include Australia (AUS), Boreal Eurasia (BEUR), Boreal North America (BNA), China (CHN), India (IND), Europe (EUR), Temperate Eurasia (TEUR), Temperate North America (TNA), Temperate South America (TSA), Southern Africa (SAf), North Africa (NAf), Tropical Asia (TrAS), Tropical South America (TrSA), Oceans (OCN), Tropics (Tr), Boreal (BOR), Northern Temperate (TN), and Southern Temperate (TS). The last row marked as SLE denotes the Siegel linear estimate of the linear trend in CH$_4$ emissions (Tg/yr/yr) over the 2004–2020 period.

| | AUS | BEUR | BNA | CHN | IND | EUR | TEUR | TNA | TSA | SAf | NAf | TrAS | TrSA | OCN | Tr | BOR | TN | TS |
|---|---|---|---|---|---|---|---|---|---|---|---|---|---|---|---|---|---|---|
| 2004 | 9.3 | 13.6 | 15.7 | 62.2 | 37.8 | 51.0 | 61.7 | 46.5 | 28.0 | 43.0 | 28.8 | 47.9 | 70.9 | 11.0 | 190.6 | 29.3 | 159.1 | 28.0 |
| 2005 | 9.1 | 13.5 | 20.1 | 51.0 | 37.7 | 44.1 | 57.1 | 49.7 | 29.2 | 43.8 | 31.9 | 45.8 | 71.7 | 13.2 | 193.1 | 33.6 | 150.9 | 29.2 |
| 2006 | 9.2 | 13.7 | 20.6 | 53.8 | 31.7 | 48.4 | 60.0 | 52.2 | 29.8 | 44.2 | 32.1 | 47.2 | 72.6 | 12.9 | 196.1 | 34.3 | 160.6 | 29.8 |
| 2007 | 8.7 | 13.8 | 20.3 | 69.7 | 45.8 | 45.5 | 59.1 | 50.9 | 28.7 | 44.0 | 32.5 | 47.6 | 72.8 | 13.1 | 196.9 | 34.1 | 155.5 | 28.7 |
| 2008 | 8.4 | 12.0 | 18.1 | 60.7 | 38.4 | 42.5 | 53.7 | 56.8 | 28.7 | 42.5 | 31.4 | 43.6 | 69.4 | 12.2 | 186.8 | 30.2 | 153.0 | 28.7 |
| 2009 | 10.0 | 12.9 | 15.4 | 67.7 | 34.5 | 51.8 | 54.0 | 44.9 | 30.7 | 45.7 | 31.5 | 49.0 | 75.4 | 11.4 | 201.7 | 28.3 | 150.7 | 30.7 |
| 2010 | 10.7 | 10.9 | 24.9 | 64.5 | 38.2 | 48.3 | 58.8 | 51.5 | 33.9 | 51.5 | 39.5 | 53.7 | 85.5 | 13.8 | 230.2 | 35.8 | 158.7 | 33.9 |
| 2011 | 10.9 | 8.8 | 16.1 | 59.4 | 45.1 | 43.9 | 68.0 | 37.1 | 31.2 | 44.1 | 33.3 | 53.9 | 79.8 | 12.4 | 211.0 | 24.9 | 148.9 | 31.2 |
| 2012 | 13.0 | 9.0 | 19.1 | 62.8 | 37.9 | 45.2 | 77.0 | 42.8 | 28.0 | 44.2 | 40.8 | 60.9 | 75.2 | 13.1 | 221.1 | 28.1 | 165.0 | 28.0 |
| 2013 | 10.3 | 9.7 | 17.3 | 64.2 | 42.5 | 39.3 | 69.8 | 35.8 | 30.0 | 41.8 | 36.2 | 58.3 | 81.9 | 12.3 | 218.2 | 27.0 | 144.9 | 30.0 |
| 2014 | 10.6 | 9.4 | 21.4 | 68.1 | 31.5 | 42.9 | 69.6 | 47.3 | 29.9 | 42.9 | 36.4 | 61.0 | 94.0 | 12.7 | 234.2 | 30.8 | 159.7 | 29.9 |
| 2015 | 10.3 | 11.3 | 18.0 | 70.9 | 33.4 | 44.4 | 67.7 | 41.0 | 33.3 | 45.3 | 43.2 | 61.6 | 89.6 | 13.9 | 239.6 | 29.3 | 153.1 | 33.3 |
| 2016 | 10.6 | 10.0 | 18.2 | 73.6 | 39.9 | 41.9 | 73.5 | 35.7 | 30.7 | 45.8 | 39.3 | 65.0 | 95.7 | 14.2 | 245.7 | 28.2 | 151.1 | 30.7 |
| 2017 | 10.4 | 10.0 | 17.1 | 68.9 | 37.7 | 44.9 | 67.5 | 40.1 | 33.0 | 45.9 | 40.8 | 62.3 | 87.1 | 13.9 | 236.1 | 27.1 | 152.4 | 33.0 |
| 2018 | 11.0 | 11.1 | 16.7 | 74.1 | 37.3 | 46.2 | 73.1 | 38.3 | 32.8 | 46.7 | 43.1 | 65.7 | 82.5 | 14.2 | 237.9 | 27.7 | 157.6 | 32.8 |
| 2019 | 12.7 | 12.1 | 16.5 | 76.4 | 38.4 | 37.6 | 75.1 | 45.4 | 29.0 | 42.5 | 42.4 | 63.8 | 86.1 | 12.9 | 234.8 | 28.6 | 158.0 | 29.0 |
| 2020 | 11.2 | 10.6 | 19.7 | 80.2 | 43.5 | 49.3 | 72.7 | 42.9 | 29.6 | 45.8 | 53.9 | 62.5 | 84.1 | 14.2 | 246.2 | 30.3 | 164.9 | 29.6 |
| SLE | 0.1 | -0.1 | -0.2 | 1.6 | 0.0 | -0.4 | 1.0 | -0.7 | 0.2 | 0.1 | 1.0 | 1.5 | 1.2 | 0.1 | 3.6 | -0.2 | 0.03 | 0.2 |

**Table A1.** Kinetic Isotope Effects (KIEs) for different isotopologues reacting with the three main sinks of $CH_4$ (OH, Cl, soil) at 298 K. A KIE indicates relative reaction rate compared with $^{12}CH_4$; the reaction rate constant is applied to the OH and Cl sinks and is dependent on temperature (T); and the scaling factor is applied to the soil sink at each timestep (handled as a negative emission).

| Isotopologue | Sink | KIE | Reaction Rate Constant | Scaling Factor | Literature Source |
|---|---|---|---|---|---|
| $^{12}CH_4$ | OH | 1 | $2.45 \times 10^{-12} \times e^{\frac{-1775}{T}}$ | n/a | Burkholder et al., 2019 |
| $^{12}CH_4$ | Cl | 1 | $9.600 \times 10^{-12} \times e^{\frac{-1360}{T}}$ | n/a | Kirschke et al., 2013 |
| $^{12}CH_4$ | soil | n/a | n/a | 1 | Snover and Quay, 2000 |
| $^{13}CH_4$ | OH | 1.0039 | $2.44 \times 10^{-12} \times e^{\frac{-1775}{T}}$ | n/a | Burkholder et al., 2019 |
| $^{13}CH_4$ | Cl | 1.06 | $9.057 \times 10^{-12} \times e^{\frac{-1360}{T}}$ | n/a | Feilberg et al., 2005 |
| $^{13}CH_4$ | soil | n/a | n/a | 1.0670 | Snover and Quay, 2000 |

**Table A2.** Sites that are included in the *in situ* inversions. All sites are part of the NOAA network, other than KRS, which is part of the JR-STATION network, monitored by NIES Japan.

| Code | Full Name | Latitude | Longitude |
|------|-----------|----------|-----------|
| ALT | Alert Station | 82.28 | -62.30 |
| ZEP | Ny-Alesund, Svalbard | 78.90 | 11.89 |
| SUM | Summit, Greenland | 72.60 | -38.42 |
| BRW | Barrow Station | 71.32 | 156.61 |
| ICE | Storhofdi,Iceland | 63.40 | -20.29 |
| KRS | Karasevoe, Siberia | 58.14 | 82.25 |
| MHD | Mace Head, Ireland | 53.33 | -9.90 |
| SHM | Shemya Island, Alaska | 52.71 | 174.12 |
| UUM | Ulaan Uul, Mongolia | 44.45 | 111.09 |
| NWR | Niwot Ridge, Colorado | 40.05 | -105.59 |
| UTA | Wendover, Utah | 39.90 | -113.72 |
| WLG | Mt. Waliguan, China | 36.29 | 100.90 |
| BMW | Bermuda | 32.26 | -64.88 |
| WIS | Ketura, Israel | 29.96 | 35.06 |
| IZO | Izana, Tenerife | 28.31 | -16.50 |
| MID | Midway Islands | 28.22 | -177.37 |
| KEY | Key Biscane, Florida | 25.67 | -80.16 |
| ASK | Assekrem, Algeria | 23.26 | 5.63 |
| KUM | Cape Kumukahi, Hawaii | 19.56 | -154.89 |
| MLO | Mauna Loa, Hawaii | 19.54 | -155.58 |
| RPB | Ragged Point, Barbados | 13.17 | -59.43 |
| SEY | Mahe Island, Seychelles | -4.68 | 55.53 |
| ASC | Ascension Island | -7.97 | -14.40 |
| SMO | American Samoa | -14.25 | -170.56 |
| CGO | Cape Grim | -40.68 | 144.69 |
| BHD | Baring Head | -41.40 | 174.87 |
| CRZ | Crozet Island | -46.43 | 51.85 |
| USH | Ushuaia, Argentina | -54.84 | -68.31 |
| PSA | Palmer Station, Antarctica | -64.77 | -64.05 |
| SYO | Syowa Station, Antarctica | -69.01 | 39.59 |
| SPO | South Pole, Antarctica | -89.98 | -24.8 |