# Peer review of "Atmospheric data support a multi-decadal shift in the global methane budget towards natural tropical emissions"

_Atmospheric Chemistry and Physics, 2022_

## Author Comment (AC1)

Interactive comments on "Atmospheric data support a multi-decadal shift in the global methane budget towards natural tropical emissions"

Drinkwater et al

We thank both reviewers for their useful comments (denoted in black italics). We provide a response to each comment.

**General comments**

This paper investigates the post-2007 global increase of methane in the atmosphere. It presents two Bayesian invers methods to estimate both, regional CH4 emissions and its  $\delta^{13}$ C signatures. Moreover, it examines the impact of a global OH reduction on the emission estimates.

The applied methods and the outcome of the study is of interest to the scientific community and fits within the scope of the ACP journal. The paper provides interesting findings and is in general well written in terms of language. Unfortunately, the paper is poorly structured and often lacks precision. I suggest to accept the paper for publication with major revision, which is mainly due to the poor introduction and missing explanations of what is being studied.

**Specific comments**

**Abstract**

The abstract is short and concise, and contrary to the introduction (see comments below), provides a comprehensive overview of this paper and its results.

**1.** Page (P) 1 Line (L) 1: "...a Maximum A Posteriori inverse method...". Please also mention the second inverse method, i.e., the Ensemble Kalman Filter.

**We have amended the text (line 1).**

**Introduction**

The introduction needs to be better structured. It lacks precise explanations (especially regarding the application of stable isotopes) and exact numbers. It should be clearly explained what is being studied and why stable isotopes are being used. I suggest to put more afford into this first part (see specific comments below).

**2.** P1 L16-18: Here the authors only explain, that  $CH_4$  is a GHG. I suggest to briefly state why  $CH_4$  is an important greenhouse gas (e.g. compared to  $CO_2$ , secondary/indirect radiative forcing) and include references. We have amended the text to reflect the importance of CH4 relative to CO2 by describing GWP values (lines 22-25).

**3.** P1 L18-19: Be careful with the terms "anthropogenic" and "biogenic". Biogenic CH4 emissions are also anthropogenic (e.g. emissions from livestock, rice cultivation, landfills, etc.). There are also non-biogenic natural emissions (e.g. from volcanic eruptions, wild

fires). Better: use "biogenic", "thermogenic" and "pyrogenic" and further distinguish between "natural" and "anthropogenic". This mistake (and the associated misinterpretation of the results) is found throughout the paper. Please check the entire text.

**We have clarified these differences throughout the paper.**

**4.** P1 L19: Please mention that the oxidation by OH is the main loss of atmospheric CH4. This explains why later in your study you only examine the role of isotopic fractionation by OH oxidation and neglect all other loss processes. Moreover, the adjective "small" in front of "stratospheric loss" is not informative, since you do not give any other orders of magnitude for CH4 loss processes.

**We have amended the text (line 21).**

**5.** P2 L21: The term "unprecedented values" is not precise. Please give exact values.

**We have added values to indicate accelerating mole fractions (line 26).**

**6.** P2 L22: What do you mean by "lighter CH4"? Please use correct terms, such as "isotopically lighter" or "depleted in the heavier stable isotope 13C". Please also indicate, if you mean the isotopic signature of globally averaged atmospheric CH4. Again, please be more precise here.

Note: The isotopic signature  $\delta^{13}$ C should also be explained at some point in the paper: why do you use isotopic signatures and what can we infer from them? In addition, please mention to which standard the carbon isotopes are reported to (i.e. relative to the Vienna Pee Dee Belemnite standard). This can also be done in the Section 2. I also suggest noting once, that throughout the text  $\delta^{13}$ C refers to  $\delta^{13}$ C in CH4.

**We have clearifed the use of lighter CH4 throughout and have detailed the background to isotope ratio data in the description of the data (lines 79-82).**

**7.** P2 L22: A change towards more negative  $\delta^{13}$ C possibly indicates a shift towards biogenic emissions. However, another explanation, could also be a reduction of isotopically heavier pyrogenic emissions. I suggest writing that this shift could indicate a change in the contributions of the emission sources.

It is true that an increasingly light signature could be an increase in biogenic emissions or a reduction of pyrogenic. We have adjusted the text to reflect possible other contributions (line 28). In terms of our main results, considering the context of the geographic location of emissions, with the emissions being centered in tropical regions such as North Africa and tropical South America, and other studies which indicate increasing biogenic emissions from these regions (for example from Lunt et al, 2019, 2021; Wilson et al 2021), we consider biogenic emissions to be the most likely cause of increasing emissions (detailed throughout, e.g. lines 360-364).

**8.** Moreover, since you also investigate the role of OH oxidation, please explain the influence of isotopic fractionation during the sink processes in the atmosphere. How does the isotopic fractionation change the  $\delta^{13}$ C signature of atmospheric CH4? We have clarified our explanation of this in the text "Reactions responsible for removing methane from

the atmosphere are faster for lighter isotopologues of CH4. This isotopic fractionation therefore leads to an atmosphere enriched in heavier isotopes relative to the globally emitted methane." (lines 48-50)

**9.** P2 L24: Explain why you start to investigate emissions from 2004 onwards. Until which year? We chose that period to capture the growth rate of CH4 from 2007. The growth rate of atmospheric CH4 was close to zero from 2000-2007, and so starting from 2004 captures some of that zero-growth period and the subsequent uptick in the CH4 growth rate.

**10.** P2 L24: What means "short-term" in this case? **This refers to monthly variations.**

**11.** P2 L27: Again: anthropogenic emissions can also be microbial. Please describe which emissions you mean exactly, I guess natural (tropical wetland) emissions. This shift described is from northern hemisphere energy emissions to biogenic tropical wetland emissions, however we have adjusted the text to reflect possible other contributions (line 29).

**12.** P2 L27-28: Does this sentence describe the motivation of your study or the conclusion from your results? Rewrite it to fit the context of the introduction or move it to the end of your conclusion or outlook. Moreover, please add a reference, if this shift "from anthropogenic to microbial sources" is not your conclusion. We have amended the text to fit the introduction to give more context to our aims and results (lines 57-60).

**13.** P2 L35. It is not explained, what the authors mean by "From the source signature of a region". Please use correct and precise terms, i.e. "the isotopic source signature" and check throughout the entire text. We have amended the text to reflect that `source signature' is always referring to `isotopic source signature' throughout the text.

**14.** P2 L38-39: Why are changes in OH unlikely to play a dominant role? I also suggest mentioning that the role of OH is highly uncertain as atmospheric OH is very difficult to quantify. The high uncertainty in OH levels (due to its short atmospheric lifetime and subsequent difficulty in measurement) means it is difficult to quantify whether it has had an impact upon CH4. We have amended the text to reflect this, describing that it 'may have' played a role, rather than being 'unlikely' (line 46).

**15.** P2 L41: Please write "isotopic fractionation" instead of "fractionation". We have clarified this along with the isotopic source signature descriptions throughout.

**16.** P2 L41-42: Do you explore the impact on the  $\delta^{13}$ C signature? If so, please add this information to your sentence.

We are looking at trends in changes in regional isotopic source signatures. We appreciate that the absolute isotopic source signatures are highly uncertain (we now directly acknowledge this in text), however, given that this work studies the trends rather than absolute source signatures we do not consider our conclusions susceptible to uncertainty in the source signatures. **17.** P2 L46: How can this be corroborated by  $\delta^{13}$ C studies? How do you investigate the assumption of reduced OH in 2020, due to lower NOx emissions?  $\delta^{13}$ C would be sensitive to a bulk reduction in OH as the fractionating power of the atmosphere would be reduced, leading to heavier atmospheric  $\delta^{13}$ C signatures. We examine these variations in a sensitivity test - for 2020, the size of the OH was reduced by 5%. This was based upon global mean reductions in OH for the first half of 2020 from **Miyazaki** et al. (2021) and Laughner et al. (2021). The impact of this change upon the regional emissions was examined in a sensitivity test (described in the `Sensitivity of Results to Changes in Assumed OH Distributions' section and results on page 10, lines 315-323), to examine the impact of reduced OH on emissions in 2020, finding that emissions are generally still within uncertainties of the test with no interannual variation in OH. As part of the peer-review process, Feng et al (2022) have formally inferred changes in OH and methane emissions and found that the decrease in OH in 2020 describes about 30% of the elevated atmospheric growth rate in that year relative to 2019.

**18.** P2 L49: Here you only mention OH, although this is just a part of your analysis. Please write exactly what else you analyse and present in Section 3. We have clarified this in the text, including all the tests performed (lines 61-64).

**19.** Up to this point, it is not clear to me what exactly the scientific questions are, and how the authors intend to investigate them. I suggest to restructure the introduction and work out or include:

- what the scientific questions are (e.g. reason of CH₄ short-term variations after 2004, the role of OH reduction, CH4 increase in 2020 due to reduce NOx emissions)
- that you are using two different approaches of inverse modelling
- that you perform these inversions for  $CH_4$  emission fluxes and  $\delta^{13}C$  signatures
- that you use in-situ and satellite-derived remote sensing observations for the inversion
- indicate if you only focus on certain regions (i.e. low, mid or high latitudes?) or on all
- the usage of stable isotopes here.

**We have added a paragraph to this section, including the key scientific questions and the approaches we have taken (lines 56-60).**

**Data and Methods**

In general, it would be easier to follow this section if the methods were briefly introduced (only 1-2 sentences, see comments above), and the need for observations to constrain emissions of different regions was pointed out. Moreover, Section 2 lacks a description of the sensitivity simulations performed to investigate the role of OH (although this is done in the last paragraph of Section 3). I suggest to restructure the paper and describe the simulations in Section2, including what is meant by "control calculation".

**20.** P3 L52: What do you mean by "total" CH4 emissions? Total emissions in the respective regions you study or total global emissions? **`Total' in this case refers regional CH4 emissions, which we have clarified throughout the text by referring to regional rather than total.**

**21.** P3 L52: Please make clear that you mean  $\delta^{13}$ C of the CH4 emissions, if you have not yet in the introduction (see also comment 7 on P2 L22). We have clarified this (line 57).

**22.** P3 L61: Is there a constant offset due to the different heights where measurements were performed? The offset is not due to different sampling heights (where measurements are co-located this will not have a measurable bias). The offset was based upon lab intercomparisons from the WMO, which reported a factor of 0.997 between the NIES 94 and NOAA 04 scales. This was corroborated by more recent intercomparisons by the WMO

(https://gml.noaa.gov/ccgg/wmorr/wmorr\_results.php?rr=rr6&param=ch4&group=gro up5)

**23.** P3 L63: Here you start to describe  $\delta^{13}$ C measurements, although you have introduced them before (L62). From the text above, it was not clear that you only described the CH4 observations. Please be more precise above. **The prior section was describing CH**4 **amount fractions, which is what was meant by total CH**4. We have amended the text to clarify this (line 67-78).

**24.** P3 L65: Please specify to which standard carbon isotopes are reported (i.e. VPDB). **These isotope ratios are reported on the VPDB isotope scale, which we have clarified in-text (line 80).**

**25.** P3 L78: I am not sure where the uncertainty of 1.2 % comes from. **This uncertainty** was chosen to account for model errors such as atmospheric chemistry and transport – these uncertainties are detailed in Feng et al., 2022. We have now detailed this in the corrected text (line 100).

**26.** P3 L80: Please give more details about the GLOBALVIEW data. Are they in-situ or remote sensing measurements, or both? In line 81 you mention uncertainties of in-situ measurement. I suggest to move the term "in situ" to the first part of the sentence (i.e. "GLOBALVIEW in-situ CH4 and CO2 data …") and also indicate if data are collected at surface level? **GLOBALVIEW are a combination of CH**4 data from ground-based data (both flask and continuous) and aircraft data, from 54 different laboratories, combined and published by NOAA-GML. We have detailed this in-text (lines 101-105).

**27.** P4 L86: How is this "driven"? We have adjusted the text here to clarify that this is the meteorological data used in the modelling (line 109).

**28.** P4 L88: Why do you use EDGAR v4.3.2, which provides only data until 2012? The latest EDGAR emission inventories provide emission estimates until 2015 (v5) and 2018 (v6). Do you repeat emissions after 2012 annually? **EDGAR v4.3.2 was chosen due to it being the most recent version when modelling work commenced. The anthropogenic emissions repeat from 2012. Fire and wetland emissions are repeated from 2015**

**onwards (this can be seen in the prior emissions in Figure A1). We have updated the text (line 116).**

**29.** P4 L90: Are agricultural waste burning emissions included in the GFED or EDGAR inventory? Please give information until which year emission estimates of the respective inventories are available. **Biofuel emissions (which covers waste burning) are obtained from the EDGAR inventories. We have detailed this in-text (line 113).**

**30.** Table 1: This table shows the global methane emissions from Saunois et al. 2016. Why are these emissions shown, although not applied to any inversion? Is there any reference in the main text? Moreover, there is a more recent publication (i.e. Saunois et al 2020), which provides emission estimates until 2017. Would it be possible to add the annual mean a priori emissions of each sector to the table? Maybe the authors could also compare their results with the emission estimates provided by Saunois et al 2020. The Saunois data is displayed to give context to size of each emissions type typically. We have updated the text to refer to it directly (line 40), and updated the values to reflect the latest version of the paper (Table 1).

**31.** P4 L95-96: In my opinion, the single example "(e.g. coal mines)" does not reflect the many different isotopic signatures within the individual sectors. Either skip the example here and discuss the different isotopic source signatures later, or give more examples. (e.g. wetland and pyrogenic emissions) and find references. The coal mines example has been removed.

**32.** P4 L99: If you mention the isotopologues here, please briefly describe why you need to convert isotopic ratios into isotopologues. Bulk isotope ratios are converted into isotopologue values because ACTMs do not model ratios but the isotopologues that make up the isotope ratio; we simulate the isotopologues separately in GEOS-Chem, then combine them into the reported bulk isotope ratios after the simulation. We detail this in the Appendix.

**33.** P4 L102-103: Is the atomic chlorine 3D field also derived from the full-chemistry simulation? Please also indicate the time period for which these 3D fields are available. **The chlorine was originally derived from a full-chemistry simulation. It represents a single year which is repeated. We have detailed this in-text (line 132).**

**34.** P4 L104: What about transport into soils, is this considered? **The soil sink of CH4 is included and we describe the sink in this section (line 129).**

**35.** P4 L105: Please also compare the 9.73 years to other  $CH_4$  lifetimes reported in the literature. This value compares well with multi-model simulations such as Voulgarakis et al (2013) and Morgenstern et al (2017), who listed the global mean lifetime of  $CH_4$  as and 9.6 ± 1.8 yr and 7.2-10.1 yrs, respectively. We have now detailed this in-text (lines 131-132).

**36.** P4 L113: From which year is this restart file? When do you start your simulation? What is the coarser resolution of the previous GEOS-Chem model simulation? **The restart file is at 4x5 degree resolution. The restart file originally represented 2012, however it is**

spun up over 60 years to the point where it no longer resembles 2012 and will be representative of 2004, the starting point of the simulation. We have clarified this intext (line 140).

**37.** P5 L143-144: Please explain the basis on which you selected these uncertainty values. The uncertainties chosen were based upon other uncertainty values for similar MAP studies, such as Fraser et al 2014 and McNorton et al 2016. We have detailed this intext (line 174).

**38.** P6 L158-159: Do I understand this correctly: Each source signature of each sector with the respective region is perturbed by 20 ‰? Please rephrase the sentence to make this clear to the reader. **Yes, the sources within each region are shifted heavier by a constant amount. We have amended the text to make this clearer (line 189-190).**

**39.** P6 L166: Are you performing this a posteriori simulation using the improved CH4 flux estimates and  $\delta^{13}$ C, for further analyses? If yes, what do you intend to analyze with this a posteriori simulation and for which time period is this simulation performed? I am not sure if I am understanding this correctly. The a posteriori simulation is used to check how it corresponds with in-situ data, to ensure the a posteriori gives better agreement with in situ data than the a priori. We have expanded this in-text (lines 197-198).

**Results**

In this section, the results are well presented. The figures are comprehensive and qualitatively well presented (with the exception of Figure 4, see comment 50). However, an in-depth interpretation is sometimes missing. I suggest discussing the results here in detail and renaming this section to "Results and Discussion".

**40.** The authors often use different and ambiguous terms to distinguish between the observational data. For example, in:

- Figure 2: the terms "in situ" and "ground-based" are used to describe the same data in only one figure caption.
- Figure 4: the terms "NOAA" and "δ13C" are used for CH₄ and isotope measurements, respectively, although δ13C data are also measured at NOAA sites.

Please ensure that you are using consistent terms for the observational data throughout the text and in the figures and figure captions. We have ensured the difference between the data types is clear in the text and all figure captions.

**41.** In the results section, the authors often refer to figures in the supplements. This direct reference makes me search for them in the text. I would suggest moving some plots that are directly described to the main text. And/or take the focus away from the figures, which remain in the supplements, e.g. by using phrases like " (see Supplements for absolute emission values)" on P7 L186 or "(described in more detail in the Supplements)". Moreover, the order in which figures are numbered does not correspond to the order in the text. We have checked and adjusted the order of the figures and have changed the wording to

**focus more on the main text figures. Specifically, we have moved Figures A1, A2 and A3 into the main text (in the new version, they are Figures 3, 6 and 7, respectively).**

**42.** P7 L187: As already mentioned above (see comment 40) please be consistent with naming of the observational data. Here only the term "NOAA" is used. What about the measuring site in Siberia (NIES), are data from this site included? I suggest to define appropriate names for both data sets in Section 2 and stick to them throughout the text. As above, we have clarified the differences.

**43.** P7 L191-192: The authors describe the emissions of the year 2020. But how do emissions evolve over the time period (2004-2020) investigated here? Emissions in Europe in 2020 have increased, too. What could be the reason? What happened in 2019 in Europe? We have chosen to focus on the year 2020 in this section due to particular interest in this year, in the context of the pandemic; this has been detailed (line 231). The increased European emissions are seen in the ground-based inversion but not from GOSAT inversion, so there is ambiguity in this result.

**44.** P7 L204: Why do we see those differences between the two results? Is there an explanation for this? There are a few reasons for possible discrepancies between the two datasets – this could be due to the relatively better geographic coverage of satellite data (limited locations of ground-based sites). Likewise, there are temporal gaps in satellite data due to the polar orbit of the satellite and instances when there is persistent cloudiness. This has been clarified in the updated text (lines 247-248).

**45.** P7 L208: Why do we see this increase in China? Other studies suggest increasing emissions from coal mining here over the last years. Please also discuss this with respect to your  $\delta^{13}$ C a posteriori results. China has a trend of isotopically lighter  $\delta^{13}$ C emissions from 2012, like most regions, however the isotopically light shift is not as strong in its corresponding latitudinal band as others (Figure 5), which suggests that heavier sources such as coal mines could be part of the emissions makeup here. However, this result is ambiguous as other emissions sources could produce the same result, such as a less significant increase in wetland emissions. This discussion is also now detailed (lines 254-257).

**46.** P8 L211-217: Please name the key message from those results, i.e. statistical analyses show improvement of the emission estimates compared to a priori emission estimates. **We have added this description in the text (line 258).**

**47.** P8 L219ff: In Figure A7 it can be seen, that the inverse modelling reduced the uncertainty of the  $\delta^{13}$ C signatures in each region. Moreover, the resulting  $\delta^{13}$ C signatures and their seasonal variability correspond to the sources which are expected in the respective regions. This is nicely explained by the authors. However, it could be described in combination with Figure 3, since here the main outcome of the inverse study is shown, e.g. the changes in  $\delta^{13}$ C over time. Moreover, the focus would not be on Figure A7, which is only shown in the Supplements (see also comment 41). We have amended the text to make the main focus be on Figure 3 with reference to A7 rather than the focus being on a supplementary figure (lines 268 onwards).

48. P8 P235-238: Why do we know that this shift in 2012 towards heavier isotopes is due to anthropogenic activities? Which anthropogenic emissions caused this shift? The author could for example support their conclusions with other studies, such as ethane measurements. Is there another explanation for this? Why couldn't it be caused by enhanced biomass burning? The suggestion of a heavy isotopic shift in 2012 being a result of anthropogenic emissions was based upon geography with the Northern Hemisphere being more responsible for anthropogenic emissions, considering Nisbet et al (2016), who determine these variations are a result of emissions changes (line 284).

**49.** P8 L238-239: The term "heavy" or "light" trend does not sound correct to me. I suggest using "trend towards lighter isotopes" or "trend towards more negative  $\delta^{13}$ C" v.v. Why is this trend "light" in 2012? In which regions do we see the "heavy" trend in 2008? Why? We have amended the text to account for this, the trends were incorrectly described (line 283-286).

**50.** Figure 4:

- Please do not assign axis labels by arrows.
- The figure caption does not say, what is shown on the right side of this plot. I suggest to divide the plot into two subplots called a) and b) and explain exactly what is being shown in subplot b).
- here the authors mix up fluxes and emission rates (which is "mass \* time-1 \* area-1" and "mass \* time-1", respectively). Please also check throughout the text if the terms "flux" and "emission rate" are used correctly.

**We have removed the arrows (now the different values are indicated by the colours of the x axes) and have expanded the figure captions. We have clarified flux vs emissions rate in the text (Figure 5 in new version).**

**51.** P9 L271ff: This part should be moved to Section 2, where the authors describe the methods and model simulations. We have moved the OH description to the methods section ('Sensitivity of Results to Changes in Assumed OH Distributions').

**52.** P9-10 L284-285: Why is the reduction only seen in high-emitting regions? **The** reduction would be seen in all regions, however, the aim of this sentence was to say that the onus will be on high-emitting regions to achieve necessary emissions reductions as expected by emissions reductions protocols (line 319).

**Conclusions**

The conclusion summarizes the paper comprehensively.

**Technical correction**

P1 L5-6: "The satellite remote sensing data provide evidence of higher spatially resolved hotspots of methane...". This sentence does not make sense to me. Do you mean: "The

higher spatially resolved satellite remote sensing data provide evidence of methane hotspots ..."? Yes, we have clarified this sentence (line 6)

P3 L65: Instead of "The geographical locations of the in-situ data..." I suggest: "The geographical locations of the in-situ measuring sites are shown in Figure 1." And maybe begin a new sentence: "They present a subset ..." However, I am not quite sure what you are trying to say with this sentence. The sites used in the inversion are not every single site in the NOAA network, just ones with entire coverage over the time period (2004-2020). We have clarified this in-text (lines 86-87).

P3 L72: Either "carbon dioxide" or "CO2". Not both. Noted, text amended (line 93)

P3 L76: "The analysis shows..." or "Analyses show..." Noted, text amended (line 97)

P4 L88: There is a missing "v": EDGAR v4.3.2 Noted, text amended (line 111)

P4 L94: Please insert "...isotopic source signatures..." Noted, text amended (line 117)

P4 L94: "which provide" or "dataset/database from Sherwood et al., which provides" **Noted**, **text amended (line 118)**

P4 L113: Please change "older" to "previous". I think the "and " is wrong here. **Noted, text** amended (line 140)

P5 L121: Please use the term "grid boxes" instead of "grid squares" (please check throughout the entire text) **Noted, text amended throughout**

P5 L121: Please change to "...correspond to the location of the sampling sites..." **Noted**, **text amended (line 181)**

P5 L131: Please be more precise and write "CH₄ fluxes" and not only "fluxes". **Noted, text** amended (line 160)

P5 L134-135: This sentence sounds complicated to me. I suggest to write: "The MAP solution and the associated a posteriori uncertainty is described as:

**FORMULA,**

using the conventional that lower-case and upper-case variables denote vectors and matrices, respectively, and where ..." **Noted, text amended (line 164-169)**

P5 L141: I think the "either" needs to me removed We have amended this to `The measurement vector includes mole fraction data OR  $\delta$ 13C data, not `and'. (line 171)

P5 L145: " $\partial y/\partial x$ " should be " $\partial y/\partial x$ " **Noted**, text amended (line 176)

P5 L148: There is a missing dot: "...normal. The individual..." **Noted, text amended (line 179)**

P6 L160:  $\delta^{13}$ C is always given in ‰ Noted, we now describe this as  $\delta^{13}$ C change instead of saying in permille (line 192).

P6 L164: "The a posteriori CH4 fluxes...". Please check throughout the text and specify which fluxes (i.e. CH4) you mean. Even though it is clear from the context, I would prefer the authors to indicate this correctly. Likewise, in L165 please write "isotopic source signatures" or " $\delta^{13}$ C source signature" and check throughout the text. We have checked and clarified the descriptions of fluxes and source signatures throughout.

P6 L165: Please add "...horizontal grid" Noted, text amended (line 196)

P8 L 228: "...assumes that sources are..." Please specify which sources. We have removed this sentence in the updated version due to moving away from focus upon appendix figures. However, this sentence was referring to the fact that we assume that seasonal cycles in  $\delta$ 13C are driven by seasonality in all emissions types, rather than sink variations.

P8 L241: Please add "atmospheric growth rate of CH₄" **This sentence has been removed** in the new version of the manuscript.

P10 L293: Here you write "post-2007 increase". In the introduction it is written "post-2006". Please be consistent. This should be post-2007 throughout, we have removed reference to post-2006 (line 32).

Table 1: Is "magnitude" in this context correct? I would prefer anything like "global mean emissions". Please also add "(‰)" to the table, as well as the uncertainty range for the isotopic source signatures. We have changed the wording of the table and added uncertainties on the isotopic source signatures (Table 1).

Table 2: I suggest to move this table to the supplements, since it does not provide any new information. **Noted, we have moved this to the supplements (Table A2)**

Figure 2A: missing space in figure caption in line 1"...GOSAT data (blue line)..." **Noted**, **text amended (Figure 6)**

Figure A3. double "n" in line 2: "..shown..." Noted, text amended (Figure 7)

**REVIEWER 2**

The paper by Drinkwater et al. studied changes in regional methane emissions and  $\delta 13C$  source signatures over the period 2004-2020, using two inversion frameworks that assimilated in-situ and GOSAT observations respectively. They found a progressively emission increase from tropical regions accompanied by lighter  $\delta 13C$  signature, and concluded a multi-decadal shift in global methane budget towards tropical natural emissions (wetland emissions notably). The subject of the paper fits in the long-term research interests in the community regarding the decadal changes in methane budgets and underlying drivers. In general, I find the paper interesting to read, and relevant to the scope of ACP. However, there are a few major concerns that may weaken the robustness of the main conclusions, and hopefully can be addressed in revisions.

One of the biggest issues is that the inversion results presented in this study lack independent evaluation. While the two inversions based on in-situ and GOSAT observations respectively do show some consistency in the overall emission trends at the global scale and large latitudinal bands (Fig. 4 & Fig. A2), there are clear discrepancies between the two inversions for big regions regarding emission increments after inversion (e.g., Boreal North America & China in Fig. 2), magnitudes of posterior emissions and emission trends (e.g., Temperate North America & Tropical Asia in Fig. A1). The good model-data agreement at some selected sites as shown in Fig. A4 is expected, as these stations were assimilated in inversions and most of them are marine boundary layer stations, where observations are normally reproduced by models. In fact, I would expect poor model performance at some difficult sites such as KRS and BKT even though they were assimilated. I suggest the authors examine model performance at all sites assimilated, and if possible, include non-assimilated sites or observations from other platforms like aircraft campaigns or TCCON sites, so as to evaluate the robustness of the inversion results for big regions.

We performed evaluation upon all the NOAA network sites that were included in the inversion, as well as at the surface-level location of all the assimilated sites from the GOSAT data. We only included a selection of site location in the figures for brevity of the paper. We have now provided independent validation by adding a figure (Figure A7) which compares additional NOAA sites that were not included in the inversion (ones that do not have coverage over the full time series 2004-2020), and still find good agreement in *a posteriori* mole fractions for these sites.

In particular, I notice that the emission trends for China since 2012 are somehow higher than the estimates from several recent papers (Lines 207–210, 0.72 Tg/yr and 1.34 Tg/yr inferred from in-situ and GOSAT data versus 0.36 Tg/yr from Sheng et al. 2021, also check out the papers by Liu et al. 2021 and Zhang et al. 2022 and references therein). The emission trend inferred from GOSAT data (1.34 Tg/yr) seems beyond the upper limit of previous estimates for the similar period, and contradicts with the recent slowdown of emission increase in China (Liu et al. 2021). Do you have any explanation for that?

**The recorded values for increases from NOAA and GOSAT data were reported in error. Recalculating the values results in values of 0.63 and 0.5 Tg/yr, respectively, which are within the range of previously reported values. We have updated the values in the text (line 253-255).**

For the optimization of  $\delta 13C$  signature, I don't quite understand the methodology. It's not clear whether regional methane fluxes and  $\delta 13C$  signatures were solved simultaneously or sequentially? According to the description of methodology in Lines 154–163, it seems that the solution of regional  $\delta 13C$  signatures relies on the solution of regional emissions. I wonder how much errors in estimates of regional emissions would impact the solution of  $\delta 13C$  signatures. *Can we trust the results presented in Fig. 3 if the emission trends detected for certain regions are not robust?*

**This comment should be covered by adding an additional paragraph of explanation to the introduction section (we cover this in the responses to reviewer 1). In the new version this detailed in lines 56-60).**

The lighter  $\delta 13C$  signature in tropical regions doesn't necessarily imply an increase in natural emissions (wetland emissions in particular). The tropical regions are known for their agricultural practices and related methane emissions, and recent studies suggested emission increase from agricultural sectors in tropical countries (Stavert et al. 2022; Zhang et al. 2022b), which could also lead to lighter  $\delta 13C$  signature according to Table 1. Is it possible that agricultural sectors also had substantial contribution to the recent trends in tropical emissions? In the abstract the authors claimed that "the satellite remote sensing data provide evidence of higher spatially resolved hotspots of methane that are consistent with the location and seasonal timing of wetland emissions" (see Lines 318–320 as well), which is not clearly shown in this paper. The authors also cited a few papers that reported large CH4 anomalies or trends in Eastern Africa or Amazon, which seems to confirm their conclusions. But it's not clear how much wetland emissions from tropical regions contributed to the signals detected in this paper.

We agree with the reviewer, and have made it clearer in the text that the evidence to support larger wetland emissions (over agriculture) comes from lines of evidence in other studies, and that our aim in this paper was to corroborate those findings through the isotope ratio measurements (lines 360-364). The hypothesis of wetland emissions being significant was based on the trend towards lighter  $\delta^{13}$ C and the fact that increasing wetlands from these regions has been suggested previously (Lunt, 2019; 2021). The trend of lighter  $\delta^{13}$ C does not conclusively mean that wetland emissions are increasing versus agriculture (or decreasing energy emissions) but was based upon other literature and the locations of the emissions increases (North Africa, Tropical South America).

The use of climatological OH fields for the reference runs is fine, given the large uncertainty in the long-term OH trends and variabilities. The authors should be aware of the range of uncertainties among recent studies (see e.g., Turner et al. 2017; Naus et al. 2021; Patra et al. 2021; Zhao et al. 2020 etc. and references therein), and discuss how this could impact methane budgets and variabilities. The paper by Lan et al. 2021 cited in the introduction (Lines 39–41) seems to deny the hypothesis proposed by Turner et al. 2017. Why did you choose decreasing OH by 0.5%/yr for the sensitivity test that followed this hypothesis? The choice of 5% uniform drop in OH for 2020 is also problematic, given the large spatial and temporal variability in OH changes in response to reduction in NOx emissions due to COVID lockdown.

The hypothesis of Turner et al (2017) was followed despite Lan et al (2021) refuting it, because this suggestion of a trend of decreasing OH has been popularly suggested as a reason for increasing CH4 amount fractions as an alternative to increasing emissions (for example McNorton et al, 2016; Rigby et al, 2017; Turner et al, 2017; McNorton et al, 2018; Fujita et al, 2020). The 5% global drop in 2020 was based upon maximum global mean values from Miyazaki et al. (2021) and Laughner et al. (2021), for the first half of 2020. This wholescale reduction does leave out smaller scale variability but is warranted, given the large uncertainties in OH amount fractions, as well as large regions and grid box size in the simulation. Feng et al 2022 performed inversions for OH in 2020 using GEOS-Chem and found that the OH sink of methane reduced by 1.4% in 2020 compared to 2019. Considering this, our reduction is greater than what has been predicted. However, the resulting emissions corresponding with a 5% decrease in OH are still generally within aposteriori uncertainty on the regional emissions. This suggests that the reduction in OH (due to emission reductions in nitrogen oxides and hydrocarbons) played only a minor role in the increases in atmospheric CH4 in 2020.

---

## Referee Report (RR1)

**Comments on acp-2022-561**

*Atmospheric data support a multi-decadal shift in the global methane budget towards natural tropical emissions.* Drinkwater et al.

Explaining why methane is rising so rapidly is important. We simply don't know what is going on. What's more, methane is simultaneously becoming relatively richer in 12C, reversing its centuries-long trend towards 13C. Why? – the simplest answer is that the rise is driven by new inputs of biogenic methane, though it is also possible that changes may be happening in the methane sins.

But this puzzle is not like most scientific puzzles. Figuring out the exact life cycle of a graptolite or the exact origins of an ancient volcanic ash in an ice core can be solved in a leisurely way. But understanding methane is urgent. It very directly affects the hopes for the UN Paris Agreement and the climate future of us all.

Drinkwater et al. make a very good attempt to address this great problem. Yes, some individual assumptions and parameter choices they make can be debated – that's what science is about – but the work is sound.

First I will list my own very minor requests for changes. Then, as requested third referee, I shall comment on the earlier assessments of the earlier version of this paper, and the responses of the authors.

***Minor comments***.

1.) Two new papers are relevant and should be considered:

Oh, Y., et al. (2022). Improved global wetland carbon isotopic signatures support post-2006 microbial methane emission increase. *Communications Earth & Environment*, *3*, 159, 1-12.

Zhang Z, et al. (2023) *Nature Climate Change*. **13**, 430–433

2.) Abstract lines 4 and 5, and in the main text conclusions - It would help general readers to have some idea of the total increase in emissions over the 17 year period – the 'acceleration' (Tg/yr/yr) is given but not the total change (i.e. how much greater emissions were in 2020 than in 2004.). That should be given in Table 1 perhaps, and mentioned in the concluding section 4. Indeed, what exactly does 'Annual Mean Emissions (This study)' convey in Table 1? Maybe that's because it's being compared with Saunois et al, but it's like saying your speed as you accelerate down a freeway is some mean between when you entered from the junction and now when you're whizzing along, foot flat on the pedal.

3.) Line 125 – maybe some discussion of Oh et al 2022 would be useful?

4.) Line 139 – Any thoughts on the OH KIE puzzle? Cantrell? Sauressig?

5.) Line 334 – Several other recent papers have also come to fairly robust conclusions that OH, while important, is not the primary driver of growth.

6.) it would help to make Table 1 more detailed, or perhaps to create an entirely new Table 2 to list all the changes in emissions and growth rates over the study years. (see comment above on the Abstract).

**Comments on the authors' responses to earlier remarks**

***Referee 1*** comments on 1. the need to assess both the robustness and weaknesses of the inversion; 2. is concerned about regional isotopic signatures; 3. is worried about the sparseness of the observational network and thus the sensitivity of the optimised fluxes to the priors; and 4. is concerned about OH.

The authors have responded with significant revisions and perhaps a softening of their conclusions, that their "results are *consistent* with result studies that have highlighted a growing role for wetland emissions".

As third referee, I agree with the good points raised by Ref 1 over the initial submission, but I also consider the authors have responded well to the comments and have made appropriate revisions. The methane problem is unconstrained – we have too few real data, whether in the measurement network or in the source signature, so we have to do the best we can. We can't put the problem off for a decade until we get more stations and better measurements.

*Referee 2* also makes helpful comments.
1. Question about 2020. This year was extraordinary for methane. So it's well worth detailed attention. Note that 2021 was also extreme. Although covid obviously had impacts on air chemistry, these dramatic growth events in 2020 and 2021 were probably not primarily because of covid. Factors like the unusual triple dip La Nina and the behavior of the Indian Ocean Dipole were surely more significant. Indeed, if the growth in 2007-2018 was interesting, the changes since 2019 seem to be of a different order.
2. Table 1 – see comment 6 above on ref. 1
3. Biogenic natural vs biogenic anthropogenic. Of course rain feeds cows as well as wetlands and these two are almost indistinguishable. Note the Z. Zhang et al. (2023) revision of wetland emissions – we very badly need new real in situ observations from wetlands, especially tropical wetlands, not models.

**Conclusion**
This is an important paper that has been well debated in review, has responded well to helpful comments, and now deserves to be published, perhaps after some small further changes. The topic is important and urgent and the work is sound, as far as can be achieved given our lack of measurements, especially in the topics. This contribution needs to be published, to become part of the wider debate.

---

## Author Response (AR2)

**Responses to final comments from reviewers**

We thank the editor and the two reviewers for their patience with our addressing reviewer comments. Below we address all remaining questions/comments left from the previous reviews.

Based on reviewer comments, we have:

1) Refined the calculations so they now use EDGAR6, which provides methane emission estimates up to 2018 after which we repeat those values. We find our new posterior results are not significantly different from our previous results but nevertheless they have helped us to understand the sensitivity of our result to prior values after 2012.
2) Improved the balance of our reporting of the results across the entire study period.
3) De-emphasised our analysis of the impact of OH. Fundamentally, we wanted to explore the extent to which an alternative change in OH (as suggested by the peer-review literature) would change our conclusions and *not* to infer an OH distribution that was consistent with atmospheric measurements of methane and d$^{13}$C-CH$_4$. While we believe that OH changes did not have a significant ongoing influence on the growth of atmospheric methane over our study period, we cannot disprove that with our experiment. We have made that point clearly in the revised manuscript.
4) Extended the comparison between the model and data at different sites.
5) Used language with more care so the reader understands that while collective evidence supports a large role for tropical wetland emissions (as discussed in the conclusions), there is a limit to what we can say with our own analysis.
6) Generally improved our descriptions of the data and methods throughout the paper. We have explained better the estimation of isotopic source signatures.

We have also taken the opportunity to remove the original summary Figure 5. While our results remain the same – tropical emissions of CH$_4$ have increased with lighter isotopic source signatures (described in our revised Figures 2 and 5) – we found the visual impact of this figure was sensitive to the choice of the baseline period.

**Report 1**

*I would like to thank the authors for their responses to the reviewers' initial comments. They made efforts to improve the presentation of their results and precision of the text. However, their revisions and replies to my comments didn't fully address my major concerns.*

We now fully address every remaining concern.

*First, it's important to evaluate the inversion results to demonstrate the robustness and weakness. I appreciate that the authors added a supplementary figure (Figure A7) to show good model performance at several un-assimilated sites from north to south. This is useful, although it would be clearer to indicate error statistics for the prior and posterior at each site to show the improvement (the same for Figure A1 & A2). I would still suspect poor model performance at some difficult assimilated sites such as KRS*

*and BKT due to uncertainties in flux or transport, which was not addressed in the authors' reply or revision. Actually, it would be useful to also show the sites where the posterior doesn't improve much, which could help identify model's deficiency and regions with large uncertainties.*

We have now extended Figure A1 and A2 to show the prior and posterior methane mole fractions at more NOAA sites that show a variety of model fits to data. As noted by this reviewer, we have compared the model with data collected at sites where we assimilate data (Figures A1 and A2) and those that are independent of the assimilation process (Figure A7). We have also added mean error statistics for the prior and posterior at each site.

*Second, I still don't quite understand the methodology and value of the regional isotopic source signature inversion. According to the description in the inversion methods, the control run of $\delta13C$ simulation was built on the posterior emissions from CH4 inversion, which to my understanding were the optimised total CH4 emissions for each region, not emissions for different sectors. Then how did you calculate the regional source signature? Given the large uncertainty of posterior CH4 emissions (as the authors admitted in the paper and responses), how much this could propagate and impact the trends in regional source signature inferred from $\delta13C$? The authors didn't address my previous comment about this in their revisions [reproduced below]. Moreover, I wonder why the authors chose to solve regional source signature rather than CH4 sectorial emissions. Although the inferred trends towards lighter $\delta13C$ in the tropics partially corroborated previous studies that reported recent growth in tropical wetland emissions, it didn't provide direct evidence to support this as the contribution from wetlands and other biogenic emissions (such as those from agricultural practices) cannot be differentiated in your inversion.*

*For the optimization of δ13C signature, I don't quite understand the methodology. It's not clear whether regional methane fluxes and δ13C signatures were solved simultaneously or sequentially? According to the description of methodology in Lines 154–163, it seems that the solution of regional δ13C signatures relies on the solution of regional emissions. I wonder how much errors in estimates of regional emissions would impact the solution of δ13C signatures. Can we trust the results presented in Fig. 3 if the emission trends detected for certain regions are not robust?*

The overarching aim of our study was to understand if three separate lines of observational evidence (remote sensing from satellite, in situ mixing ratios and isotope ratios) support a hypothesis on the long-term trends driving the global methane rise. Our approach has therefore purposely not combined all evidence into a single inversion, and we consider the satellite and in situ mixing ratio inversions as the primary lines of evidence for the study.

To address the reviewer comment about whether the isotopic source signatures can be trusted given their reliance on the posterior emissions estimates, we have now edited the text to bring the reviewer's point into the discussion and to make it clear that we interpret the isotopic data as a supplementary line of evidence that is in support of the emissions estimates calculated from the CH$_4$ inversions. It is well understood that regional isotopic source signatures are very poorly constrained owing to the

uncertainty on the source signatures of individual sectors and how they vary over space and time. The isotope ratio measurements are also sparse relative to the mixing ratio time series. We therefore decided to remove the isotopic data from the first inversion that solves for the regional total $CH_4$ emissions, which prevented potentially inaccurately assigned isotopic source signatures on influencing the trends in the methane emissions estimation.

By keeping the isotopic evidence separate we can also analyse those results independently (as we have done for remote sensing vs in situ mixing ratios) although only in context of prescribed emissions estimates (as highlighted by the reviewer). We have improved the description of the methodology to communicate our approach and the reasons for it more clearly.

*The use of repeated prior emissions after 2012 could also be problematic. As the observation network are sparse globally, the optimised fluxes would be sensitive to the choice of the priors. The use of updated inventories (e.g., EDGARv5 or EDGARv6 over EDGARv4.3.2) are encouraged as errors in earlier versions were corrected in new versions. In Figure A4, the spikes in the emission signatures of Tropical Asia and South America disappeared in the later period of your simulation, probably due to the use of repeated fire emissions after 2012. Have you evaluated the impact of using repeated priors after 2012 on your inversion results? The authors claimed that the trends towards isotopically lighter sources was started since 2012 (Figure 4 and the corresponding text), could it be somehow related to the specific configuration of the priors?*

This has now been addressed. We have used EDGARv6 that describes emissions from 2004 to 2018, after which we repeat 2018 emission estimates. Figures A1 and A7 show that the prior inventory already does a good job after 2012, better than what we used previously. We find our posterior results are not significantly different from our previous results but nevertheless they have helped us to understand the sensitivity of our result to prior values after 2012.

*Third, the sensitivity test results using only one particular OH scenario (decreasing by 0.5%/yr for 2004-2019 and 5% for 2020) were mostly within the posterior uncertainty of the control run using the climatological OH. In my opinion, this does not imply that the variations in OH was not likely to have significant impacts on the atmospheric methane growth for the study period. It only reflects the fact that the inversion system is not well constrained (so that changing OH trends and variabilities doesn't change the inversion results too much). The authors should do better to prove this.*

We have de-emphasized our OH experiment, as suggested by this reviewer. We wanted to explore the extent to which an alternative change in OH (as suggested by the peer-review literature) would change our conclusions. While we believe that OH changes did not have a significant ongoing influence on the growth of atmospheric methane over our study period, we cannot disprove that with our experiment.

We agree with this reviewer that because our inversion does NOT include OH in the state vector so we cannot comment on the ability of the *in situ* data to estimate

methane emissions and OH. This involves more in-depth calculations that are outside of the paper. However, recent work by Feng et al, 2023 showed that GOSAT and *in situ* data together can independently constrain estimated geographical changes in OH and methane emissions on coarse spatial regions.

*Overall, the methodology and results presented in this paper are not adequate to support the main conclusions (i.e., multi-decadal shift in the global methane budget towards natural tropical emissions, particularly wetland emissions; limited role of OH variations in atmospheric methane growth), despite some consistency with former studies claimed by the authors. In terms of writing and the presentation of results, the revised version has been improved, but still lack precision and details.*

We have gone through the paper and improved the text for detail and precision. These changes can be seen in the tracked changes version of the manuscript.

For clarity, we say that our results are *consistent* with result studies that have highlighted a growing role for wetland emissions. We agree with the reviewer that we cannot say anything more with the data that we have reported in this manuscript. In the revised conclusions, we have attempted to bring together and interpret the different lines of empirical evidence.

**Report 2**

*Overall, the manuscript's structure has improved, and the introduction now sets out more clearly what is being studied and why. However, I have a few major and minor comments on the current version.*

Below, we describe how we systematically address all the remaining comments from this reviewer.

Major comments:
*1. P8 L 228 ff: The results part only discusses emission changes in 2020. It is still not clear to me why the authors are interested in the COVID-19 lockdown year 2020, instead of evaluating the whole period 2004-2020 and the trend of emissions after 2007. Is this not the actual purpose of the study? In my opinion, these are two different scientific questions (i.e. 1. the changes in CH4 source contribution after 2007, 2. changes in CH4 emissions/OH during the COVID-19 lockdown).*

We agree with this viewpoint. The focus on this study was not the recent changes by the longer-term change since 2004. We now briefly discussion 2020 but since this has now been addressed by a few papers so we can put our study into context of recent findings.

*However, the sensitivity study considering a 5 % drop in the OH concentration in 2020 testing the robustness of the presented results is fine. Although I miss explanations on the results (see my previous comment 52). A discussion of the OH uncertainties could also be added here (as in my previous comment 14).*

Based on this comment and the comment from Report 1 we have de-emphasised this sensitivity run but for completeness we do discuss the results we find and include a discussion of OH uncertainties.

*2. Table 1: The emissions table is shown again, without any reference to the results of this study. Can a posteriori estimates be compared to those values shown in the table? Are they within the uncertainty ranges of Saunois et al.?*

We apologize for this oversight. We have now included our a posteriori estimates into Table 1 and commented on their agreement with Saunois et al.

*3. Unlike the first version, the authors now distinguish between biogenic natural and biogenic anthropogenic emissions. A posteriori estimates indicate a shift towards more tropical biogenic emissions. In their conclusion, the authors claim that these biogenic emissions originate from wetlands. How is this conclusion derived exactly? Which satellite data do you use to identify wetland regions? This should be discussed in more detail. Would the increasing seasonality in tropical regions shown in Figure 4A support an increase in natural emissions, e.g. from wetlands?*

We agree with both reviewers that we cannot attribute the observed shift in methane emissions exclusively to an increase in tropical wetland emissions. The most we can say that the location of increased methane emissions and isotopically lighter $\delta^{13}C$ is consistent with the known locations of large wetland regions. We have now made that clearer throughout the manuscript.

Minor comments and technical corrections:
*1. Some of my comments have misleadingly resulted in these points now being described in too much detail, rather than the actual point being made briefly and more precisely. E.g.:*

*Comment 2: Instead of explaining the GWP, the sentence "CH4 is the second most important GHG in terms of anthropogenic radiative forcing" would have been fine, too.*

*Comment 6: I only missed the explanation on the usage of the isotopic signatures, such as the authors now describe on P3 in L82 ff (the exact explanation of the delta value was not necessary).*

*However, I leave it to the authors to decide how much of the newer details they want to keep in current version.*

We have now addressed these points. See tracked changes.

*2. P2 L21: delete second "from"*

Done.

*3. P2 L29: Do Lan et al. (2021) really assume a decrease in thermogenic sources?*

The reviewer is correct. Analysis reported by Lan et al (2020), including measurements of $\delta^{13}C\text{-}CH^4$, suggest that thermogenic sources are unlikely to be the dominant driver

of the post-2006 global mean increase in atmospheric methane. We have now clarified that point.

*4. P2 L51: "…OH proposed by Turner et a. (2017) are…"*

Done.

*5. P4 L116: I guess, that the inversion has no problem with upscaling yearly repeated emission of 2012. However, the a posteriori uncertainty would be larger. Could you briefly state the effect on the systematic underestimation in the a priori emissions from 2012 instead of those between 2012 to 2020?*

This is a point also raised by the other reviewer. We have now addressed this directly by using EDGARv6.0 that describes methane emission from 2004 to 2018; after 2018, we repeat 2018 emissions . We find our posterior results are not significantly different from our previous results that used 2012 emission in years later than 2012, but nevertheless it helped us to understand the sensitivity of our result to prior values after 2012.

*6. P8 L23-24: I do not understand the authors' explanation on why they only consider decreasing OH trends between 2004 and 2019. Where is the connection to an OH decline during COVID-19 in 2020?*

First, we have reduced our emphasis of changes in OH in the paper. We report the results from an idealised experiment, but nevertheless if provides us with some idea of the robustness of our results.

We argue that there is gradual, long-term geographic shift of methane emissions to the tropics that is independent of the perturbation in 2020.

The underlying explanation for the increase in atmospheric methane in 2020 is a combination of increased emissions and a decrease in OH (due to reducing emissions of nitrogen oxides from the widespread shutdown of manufacturing during Covid-19).

---

## Author Response (AR3)

**Author responses to reviewer comments on acp-2022-561**

We thank the two additional reviews of the paper. We respond below to all comments.

**REVIEWER 1**

**Comments on acp-2022-561**

*Atmospheric data support a multi-decadal shift in the global methane budget towards natural tropical emissions.* Drinkwater et al.

*Explaining why methane is rising so rapidly is important. We simply don't know what is going on. What's more, methane is simultaneously becoming relatively richer in 12C, reversing its centuries-long trend towards 13C. Why? – the simplest answer is that the rise is driven by new inputs of biogenic methane, though it is also possible that changes may be happening in the methane sins.*

*But this puzzle is not like most scientific puzzles. Figuring out the exact life cycle of a graptolite or the exact origins of an ancient volcanic ash in an ice core can be solved in a leisurely way. But understanding methane is urgent. It very directly affects the hopes for the UN Paris Agreement and the climate future of us all.*
 *Drinkwater et al. make a very good attempt to address this great problem. Yes, some individual assumptions and parameter choices they make can be debated – that's what science is about – but the work is sound.*

*First I will list my own very minor requests for changes. Then, as requested third referee, I shall comment on the earlier assessments of the earlier version of this paper, and the responses of the authors.*

***Minor comments***.

*1.) Two new papers are relevant and should be considered: Oh, Y., et al. (2022). Improved global wetland carbon isotopic signatures support post-2006 microbial methane emission increase. Communications Earth & Environment, 3, 159, 1-12. Zhang Z, et al. (2023) Nature Climate Change. **13**, 430–433*

Good suggestion by the reviewer. The Oh et al study is relevant to our discussion about the tropical wetland signature of carbon isotopes, and we have now included it in the

revised manuscript. The Zhang et al reference is less relevant for our work, although we appreciate it contains a useful model-led message.

*2.) Abstract lines 4 and 5, and in the main text conclusions – It would help general readers to have some idea of the total increase in emissions over the 17 year period – the 'acceleration' (Tg/yr/yr) is given but not the total change (i.e. how much greater emissions were in 2020 than in 2004.). That should be given in Table 1 perhaps, and mentioned in the concluding section 4. Indeed, what exactly does 'Annual Mean Emissions (This study)' convey in Table 1? Maybe that's because it's being compared with Saunois et al, but it's like saying your speed as you accelerate down a freeway is some mean between when you entered from the junction and now when you're whizzing along, foot flat on the pedal.*

We have now addressed this suggestion by tweaking the abstract and adding Table 2. For the abstract, we have stated that we use the Siegel linear estimator to determine the linear trend. With respect, we don't think reporting the emissions from 2004 and 2020 is insightful – some regions show a progressive increase with time while other regions show a large year to year variation but with an overall increase/decrease – so that reporting values for two years is not sufficient to reconcile with the overall linear trend being reported. Annual mean emissions from individual regions are reported in Figure 3. In response to this reviewer comment, we have added Table 2 that includes the information requested.

The reviewer is correct that we reported the mean statistic because it is reported by Saunois et al. We compared our results with Saunois et al, following a previous reviewer suggestion.

3.) Line 125 – maybe some discussion of Oh et al 2022 would be useful?

Agreed. Now addressed.

4.) Line 139 – Any thoughts on the OH KIE puzzle? Cantrell? Sauressig?

For the OH KIE, we used *a* value based upon Saueressig et al, which differs from values proposed by Cantrell et al. We use this value because it is the recommended value from JPL Chemical Kinetics studies, version 19 (Saueressig et al is the latest comprehensive study on the OH and $^{13}CH_4$ reaction). We understand that discrepancies between studies are not resolved and that more work is required, but a discussion is not within the scope of this paper.

5.) Line 334 – Several other recent papers have also come to fairly robust conclusions that OH, while important, is not the primary driver of growth.

Agreed. We have now contextualized our result but acknowledge (following a prompt by a previous reviewer) that our paper is not the right study to highlight the role for OH.

6.) it would help to make Table 1 more detailed, or perhaps to create an entirely new Table 2 to list all the changes in emissions and growth rates over the study years. (see comment above on the Abstract).

We have put together a new Table 2 in response to this reviewer comment.

**Comments on the authors' responses to earlier remarks**

**Referee 1** comments on 1. the need to assess both the robustness and weaknesses of the inversion; 2. is concerned about regional isotopic signatures; 3. is worried about the sparseness of the observational network and thus the sensitivity of the optimised fluxes to the priors; and 4. is concerned about OH.

The authors have responded with significant revisions and perhaps a softening of their conclusions, that their "results are consistent with result studies that have highlighted a growing role for wetland emissions".

As third referee, I agree with the good points raised by Ref 1 over the initial submission, but I also consider the authors have responded well to the comments and have made appropriate revisions. The methane problem is unconstrained – we have too few real data, whether in the measurement network or in the source signature, so we have to do the best we can. We can't put the problem off for a decade until we get more stations and better measurements.

Agreed. We appreciate this input from the reviewer.

***Referee 2*** *also makes helpful comments.*
*1. Question about 2020. This year was extraordinary for methane. So it's well worth detailed attention. Note that 2021 was also extreme. Although covid obviously had impacts on air chemistry, these dramatic growth events in 2020 and 2021 were probably not primarily because of covid. Factors like the unusual triple dip La Nina and the behavior of the Indian Ocean Dipole were surely more significant. Indeed, if the growth in 2007-2018 was interesting, the changes since 2019 seem to be of a different order.*

We resisted a detailed discussion about 2020 and 2021, given previous reviewer comments, and as this reviewer acknowledge we have softened some of our conclusions. We have added a statement in section 4 referring the reader to more detailed studies on this subject.

2. Table 1 – see comment 6 above on ref. 1

We have now included a new Table 2 including the information requested.

*3. Biogenic natural vs biogenic anthropogenic. Of course rain feeds cows as well as wetlands and these two are almost indistinguishable. Note the Z. Zhang et al. (2023) revision of wetland emissions – we very badly need new real in situ observations from wetlands, especially tropical wetlands, not models.*

Agreed. There is a possibility of a climate feedback mechanism involving ruminants, but anomalous flooding (associated with increased wetland emissions of methane) can also lead to the loss of cattle as they are swept away. There is evidence that local communities are taking advantage of larger flooded areas in S. Sudan to cultivate rice, which add the difficulties associated with robust source attribution.

**Conclusion**

*This is an important paper that has been well debated in review, has responded well to helpful comments, and now deserves to be published, perhaps after some small further changes. The topic is important and urgent and the work is sound, as far as can be achieved given our lack of measurements, especially in the topics. This contribution needs to be published, to become part of the wider debate.*

We appreciate that this reviewer appreciates the importance of our work as part of the wider debate.

**REVIEWER 2**

*The analysis use state-of-the-art techniques, although the complex inverse modelling results, using also carbon isotope signatures, are difficult to check by an reviewer. The real test would come from independent modelling trying to replicate these model results, and for this reason it is important that the authors make available the exact model assumptions, and data selections used in this modelling. My minor suggestion here refer to some clarification issues in the abstract and conclusions that make the interpretation of these results more accessible to a larger audience.*

We agree that sufficient information should be provided to the reader so they can reproduce our calculations.

*375-380 an extended discussion of the above could be included here and summarized in the summary.*

Table A2 and Figure 1 show the sites chosen for the ground-based inversion. The sites were chosen as ones that had complete monthly coverage over most of the time period examined (2004-2020). Model assumptions are described in section 2.2, including details about the isotopic simulations (Table 1). Assumptions about the inversions are described

in section 2.3. We believe there is sufficient information in the manuscript to reproduce our work.

To address this comment, we have added references to the site selection in section 4 and have linked our consistency with previous studies to confidence in our model assumptions and data selection.

*l. 8 an increased biogenic source - aka wetland source- in the tropics. Explain here what main mechanism(s) are responsible for this: temperature, rainfall, wetland extents.*

Done.

*l. 10 heavier isotope sources: what could that mean on the ground wrt source changes in China?*

Heavier isotopic signatures in China are indicative of greater proportion of sources from thermogenic or pyrogenic sources than is suggested by the prior emissions inventories.

*l. 15 what is meant with 'robust against', does it mean 'not sensitive to'? What is it that is not consistent with the global growth- and could there be an additional missing process that could make it consistent.*

We have now clarified this point. Our use of 'robust' means that the agreement is within the uncertainty of the estimate and/or the observations.

We have revised that last statement so it is now clearer. Changing OH too much is not only inconsistent with observed changes in methane but other trace gases that have OH oxidation as their dominant loss process. A detailed discussion is outside the scope of this paper, as noted by a previous reviewer.

*What is the consistency of this statement with the recent paper of Stevenson et al. https://doi.org/10.5194/acp-22-14243-2022 that suggested a strong role in 2020 for methane concentration growth? (for discussion).*

The analysis of Stevenson et al is an anomaly within the wider debate – it appears very little observational data was used to support the conclusions being reported and so it is difficult to reconcile it with a growing number of studies that have used a variety of data to test the hypothesis about a larger role for OH in determining atmospheric methane growth in 2020. In section 4, we have now pointed the reader to more detailed data-driven studies of the methane growth rate in 2020 and the role of OH.

---

## Author Response (AR4)

**Authors' response**

No additional responses are required.